# Constructing a Forest Color Palette and the Effects of the Color Patch Index on Human Eye Recognition Accuracy

Wenjing Han [1,2], Chang Zhang [1,2,*] , Cheng Wang [1,2] and Luqin Yin [1,2]

1. Research Institute of Forestry, Chinese Academy of Forestry, Beijing 100091, China
2. Key Laboratory of Tree Breeding and Cultivation and Urban Forest Research Centre, National Forestry and Grassland Administration, Beijing 100091, China
* Correspondence: zhangchang_caf@caf.ac.cn; Tel.: +86-10-6288-0719

**Abstract:** As the first visual element, color is the most attractive in the forest landscape. There are various kinds of forest colors; however, the human eye's ability to recognize them is limited. In order to combine color composition and human eye recognition ability to quantify forest colors more appropriately and to improve the ornamental effect of forest color landscapes more precisely, we have constructed a forest color palette using k-means clustering based on the color information of 986 forest images from 40 national forest parks in China. The differences in color recognition accuracy and sensitivity among populations and colors were analyzed. The effect of forest color patch indices on color identification accuracy for interior and distant forest landscapes was also explored. The results were as follows: (1) forest color could be divided into eight color families—orange, yellow, yellow-green, green, blue-green, blue, purple, and red. (2) For humans, the recognition accuracy was highest for green and lowest for blue-green. (3) For interior forest landscapes, the mean area proportion and fractal dimension of the color patches showed significant positive effects on color recognition accuracy, whereas the number and density of color patches showed significant negative effects. For distant forest landscapes, the density and Shannon's diversity index of the color patches showed significant positive effects for color recognition accuracy, whereas the number, edge density, division index, and cohesion of the color patches showed significant negative effects. We thus suggest that it is necessary to increase the complexity of the color patch shape when creating interior forest landscapes and to focus on the diversity and balance of color matching when creating distant forest landscapes. In future studies, the collection pathways for forest images should be expanded, and color information extraction algorithms that incorporate human perception should be selected. This will improve the data available for forest color studies and enable the construction of a more accurate forest color palette.

**Keywords:** forest landscape; k-means clustering; forest color palette; vision; human color recognition; color patch index; China

## 1. Introduction

Forests have varied landscapes, which attracts people to approach and enter the forest [1]. The rich colors of forest plants constitute one of the important factors of its attraction. Of the information people gather from the outside world, 87% is received through vision [2]. Color, as the first visual element, is more direct than form and size in conveying visual characteristics and stimulating human visual perception [3] and is most capable of attracting people's attention.

Forest color is derived from the changes in the growth and development of the various organs of the plant, as well as from the adaptations brought about by seasonal climate changes [4]. The composition of forest colors is so abundant that they can be precisely divided into thousands of colors [5,6]. How to reasonably quantify forest colors and construct a forest palette that responds to the actual situation remains a difficult issue for research.

The construction of a forest color palette usually requires clarifying two main issues: the selection of color space [7] and the categorization of similar colors (color threshold division [8]). There are numerous choices of color spaces, such as RGB, XYZ, CMYK, CIELAB, and HSV [9]. Of these, HSV is closer to the human perception of color [10–12], and the three components of hue (H), saturation (S), and value (V) are independent of each other [4,13,14]. The HSV color space is thus more conducive to the quantification and analysis of color and is widely used in forest color research. The current methods for categorizing similar colors are mainly divided into two categories: segmentation and clustering. The segmentation method is to divide the color space into several subspaces according to the distribution of colors and select one most representative color in each subspace; it has the advantage of short processing time [15]. Forest color studies have mostly borrowed color thresholds from image retrieval and other industries to construct a forest color palette [11,16], and there is currently no color palette available that has been designed specifically to match actual forest colors. Moreover, there are various methods to divide forest color thresholds that have not been harmonized [17]. The clustering method categorizes the colors with a higher degree of similarity until all colors are categorized [15]. There are many types of clustering algorithms, such as k-means [18], fuzzy c-mean [19], and mean shift [20]. Among these, k-means is an unsupervised algorithm with the advantages of good clustering effects and simple operation [21] and is consequently more commonly used in clustering analysis. Categorization of similar colors by clustering is widely used in research areas such as image classification, image retrieval, and image recoloring [22], but this method has been less used to study forest color. Regardless of the segmentation method or clustering method, only the physical properties of the color are quantized, without consideration of the perceptual properties of the color [23].

As viewers of the forest landscape, people gain physiological and psychological benefits from viewing actual forest scenes and viewing forest pictures [24,25]. While humans have a great ability to recognize colors and distinguish the subtle changes of colors (approximately 2.3 million colors could be perceived by human eyes in natural scenes [26]), many forest colors are ineffective or non-sensitive [27] because they lack practical significance in the process of landscape appreciation [28], physiological and psychological response. Therefore, the construction of a forest palette needs to incorporate human eye perception [29]. If so, we will be able to evaluate the visual attractiveness of forest landscapes more accurately, so as to cultivate scenic forests which meet the aesthetic needs of the public. There are differences in the human eye's ability to distinguish different colors [30], but less study corrects the human eye color perception for the constructed forest color palette. In addition, the human eye perception situation of different color space distributions varies when the color composition is the same [31]. Color spatial pattern refers to the distribution and combination regulation of color units of different sizes, shapes, and attributes in space and is the concrete expression of color landscape heterogeneity [32], which is usually quantitatively analyzed using color patch indices [16,33]. However, the relationship between color–spatial distribution and the human eye color recognition effect has not been studied.

The purpose of this study was to construct a color palette that matches the actual situation of a forest, and to clarify how different forest colors and their spatial distribution affect human eye recognition, which will be used to guide the selection and spatial configuration of colors in forest landscape creation. Located on the edge of the city or in the suburbs, forest parks provide places for sightseeing, leisure, entertainment, and science education [34]. Forest parks are dominated by forests, which can represent the basic conditions of vegetation in the area (such as plant species composition and spatial distribution) [35]. Therefore, we take forest parks as the study object, which can reflect the regional characteristics of forest color landscapes. There was no significant difference between the forest color landscape perception evaluation in the form of image stimuli and the field evaluation [36,37]. In this study, color information from forest images from Sina Weibo (an important platform for the public to share their experiences) [38,39] was

extracted to construct a forest color palette. We also designed a human eye color matching experiment to clarify the accuracy and sensitivity of human eye recognition for the developed palette and further explored the factors influencing the accuracy of forest color recognition. This study addresses the following questions: (1) How can a color palette be constructed to reflect real-world forests? (2) How do human eyes perceive the forest color palette? (3) What effect do color patch indices have on human eye recognition of forest colors?

## 2. Materials and Methods

### 2.1. Data Source

As of 2019, there are 897 national forest parks in China (https://www.maigoo.com/goomai/167110.html, accessed on 17 February 2023). We selected 30 provincial administrative regions in Mainland China (except Ningxia), and one to two national forest parks were randomly selected in each of these regions. Overall, 40 national forest parks were selected for this investigation (Figure 1, Table A1).

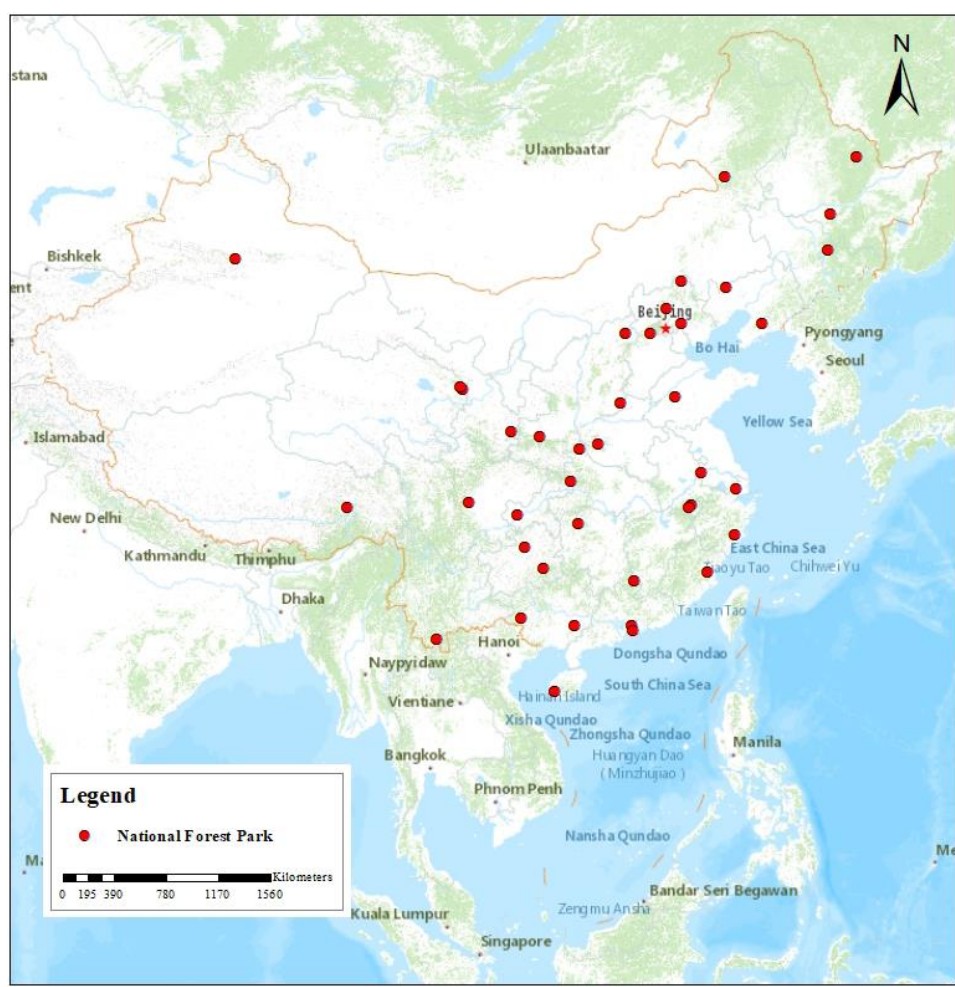

**Figure 1.** The distribution of 40 national forest parks.

The time range for the national forest park images was from 1 December 2018 to 30 November 2019. We implemented a focused crawler [40] through the Scrapy framework to grab national forest park images on Sina Weibo [41]. The specific operation was to employ the park name as the keyword, obtain the IP address data through the domain name resolution, and download the webpage images [42]. Finally, we acquired images of forest parks in different regions with a wide variety of types, including multiple viewing

distances (interior forest landscapes, distant forest landscapes) and viewing angles (flat, elevated, and overhead).

*2.2. Construction of the Forest Color Palette and Quantification of Forest Color*

2.2.1. Selection and Processing of Forest Images

Selection of forest images. We manually screened images according to the principle that the area of the forest accounted for 60% or more of the total area of the forest park images and that they represent the four seasons as much as possible. A total of 986 high-quality forest images were ultimately selected as they were consistent with the actual color situation, including 227 from North China, 224 from East China, 165 from Northwest China, 109 from South China, 101 from Southwest China, 92 from Northeast China, and 68 from Central China (Table A1).

Processing of forest images. To eliminate the influence of non-forest element colors, Adobe Photoshop CC software was used to delete the color components which are independent of the forest, such as buildings, sky, and water, to obtain forest color images consisting only of forest parts [20]. Additionally, there was the problem of non-uniform quality of forest color images due to differences in shooting equipment, shooting environment, and other factors. We uniformly processed the forest color images to reduce this problem and thus ensure the reliability of the results. (1) Unified adaptive gamma correction [43] to reduce the impact of the shooting environment (such as excessive clouds and insufficient light) on the overall brightness of the image [44]. (2) Unified automatic color equalization processing to simulate the color constancy of the human visual system and ensure the authenticity of the image color [45].

2.2.2. Main Color Extraction from the Forest Color Images

The k-means is a common unsupervised algorithm for clustering [46], with obvious clustering effects and simple operation, which is advantageous to use because of its easy implementation and high efficiency [25]. Under normal circumstances, different images have a different number of main colors [47]. When the number of colors is between three and seven, the main color features of the image can be extracted [48]. In this study, we extracted the main colors of forest color images using k-means clustering ($k \in [3, 7]$) and obtained the optimal k value using the elbow method [25]. The error squared criterion function E was used as the criterion by which to judge the clustering effect [49]:

$$E = \sum_{i=1}^{n} \min_{j \in \{1,2,\cdots,k\}} ||x_i - p_j^2|| \tag{1}$$

where, $x_i$ is the pixel point, $p_j$ is the jth initial cluster center, and $k$ is the number of clusters. The smaller the *E* value, the more compact the cluster is and the more independent the clusters are.

The algorithm process was as follows: (1) select k color samples as the initial clustering center and agglomerate the remaining color samples to the corresponding clustering according to the principle of minimum Euclidean distance; (2) use the sample mean in each cluster as the new cluster center, repeat the above steps until the cluster center no longer changes; and (3) record the RGB value of the final cluster center [44,49].

2.2.3. Construction of Forest Color Palette

The HSV color space is the closest to human color perception, making it practical to use a non-uniform quantization according to different ranges of color in the proportion of H:S:V = 8:3:3. In this study, we constructed a forest color palette based on the critical value rule (Figure A1). We first converted the RGB values for each image clustering center into an HSV value according to the formula and then conducted a secondary k-means clustering (k = 8) for hue. Finally, the range threshold of the hue was divided according to the clustering results, and the range thresholds for saturation and value were divided

by uniform quantization (divided into three levels, i.e., 1 = low [0, 0.33], 2 = medium (0.33, 0.67], and 3 = high (0.67, 1]) [50].

### 2.3. Recognition of the Forest Color Palette by the Human Eye

### 2.3.1. Experimental Design

A total of 40 experimental images were selected from the 986 forest images, depending on no interference from non-forest factors (such as sky and buildings) and the maximum coverage of all forest colors. The selection contained 5 test images to familiarize participants with the color matching process, 20 images of interior forest landscapes dominated by individual ornamental plants, and 20 images of distant forest landscapes dominated by color matching of ornamental plant communities (Tables A3 and A4). We then compiled a forest color matching program using Visual Studio 2017 software to carry out a human color matching experiment (Figure A3). The accuracy of the color recognition was measured based on the contrast between the clicked color and the actual color contained in the image, and the sensitivity of the color recognition was represented by the time taken for each color matching.

### 2.3.2. Participants

Thirty graduate students were randomly selected from the China Academy of Forestry, of which there were 15 males and 15 females ranging 18–35 years. All participants were in good health, with a normal sense of color and corrected vision above 1.0, with no additional eye problems, such as color blindness, strabismus, amblyopia, or astigmatism. The participants were informed before the experiment that their experimental data would be recorded and analyzed and that all data were treated anonymously. All participants agreed and signed a written informed statement (Figure A4).

### 2.3.3. Procedure

One day before the experiment, we recorded the participants' birth places to determine whether the participants were native (birth place accord with the image's origin place) or non-native (birth place disaccord with the image's origin place). We also sent the recorded instructional video to the participants by email, informing them of the experiment's operation. The experiment was performed in a 30 square meters laboratory. During the experiment, the interference of external factors (light, touch, noise, smell, etc.) was minimized. We closed the curtains and used artificial light to eliminate eye fatigue. The stimulus images were presented on a 23.8-inch monitor (screen resolution 1920 × 1080 pixels, 60 Hz). Participants were seated 600–650 mm from the central monitor [51] (Figure A3). The specific operation of the experiment was as follows: while carefully observing the experimental images, the participants clicked on all colors which they saw in the forest color palette according to their own color perception (Figure A3). Simultaneously, the sequence number and click time for each of the selected colors were automatically recorded.

### 2.4. Calculation of Forest Color Patch Indices

According to the constructed forest color palette and the definition of a patch from landscape ecology, we considered a relatively homogeneous nonlinear area composed of the same color as a forest color patch [52]. We completed the color analysis program using Visual Studio 2017 to interpret color information for the 40 experimental images and automatically calculated the area and perimeter of each color patch. The results were input into Excel 2016, and the patch indices of each color were calculated (Table 1, Table A5).

**Table 1.** Forest color patch indices.

| Type | Indicators | Abbreviations |
|---|---|---|
| Area | Number of color patches | NP |
| | Largest color patch proportion index | LPI |
| | The mean area proportion of color patch | ARP |
| | Color patch density | PD |
| Edge | The mean circumference of color patch | C |
| | Edge density of color patch | ED |
| Shape | Fractal dimension of color patch | FRAC |
| Aggregation | Division index of color patch | DIV |
| | Cohesion of color patch | COH |
| | Splitting index of color patch | SPL |
| Diversity | Simpson's evenness index of color patch | SIEI |
| | Shannon's diversity index of color patch | SHDI |

*2.5. Data Analysis*

All data were sorted in Excel 2016. R, version 4.1.3 (R Core Team, Vienna, Austria) was then used for analysis and mapping.

Chi-square tests were used to analyze the differences between accurate identification and inaccurate identification at the single factor level, i.e., gender, native or non-native, and interior or distant forest landscape. Statistical significance was indicated by a two-sided $p$-value $< 0.05$.

The Kruskal–Wallis test was used to analyze the differences in recognition sensitivity among the colors in the interior forest landscapes and distant forest landscapes, respectively. A data standardization method was used to avoid large differences among the values for the color patch indices and normalize all color patch indices data to [0, 1], and then the Kruskal–Wallis test was used to inspect the differences in color patch indices among the different colors and among the color families in the interior or distant forest landscapes, respectively. A value of $p < 0.05$ was considered statistically significant.

Multiple logistic regression was used to analyze the effect of each color patch index on the accuracy of color recognition. The color patch indices of the interior forest landscape and the distant forest landscape were used as independent variables, and the accuracy of color recognition was used as a dependent variable. The tolerance and variance inflation factor (VIF) was used to test collinearity among the variables, ensuring that the VIF values for all the explanatory variables were <10 [53]. Nine color patch indices were retained among the explanatory variables for color identification accuracy in the interior forest landscapes, including the number of patches (NP), density (PD), mean area proportion (ARP), mean circumference (C), edge density (ED), fractal dimension (FRAC), splitting index (SPL), Simpson's evenness index (SIEI), and Shannon's diversity index (SHDI). Ten color patch indices were retained among the explanatory variables for color recognition accuracy in distant forest landscapes, including NP, PD, ARP, division index (DIV), C, ED, FRAC, SPL, cohesion (COH), and SHDI. McFadden's Pseudo R2 was used to evaluate the goodness-of-fit of the logistic regression model [54]. The higher the value (the maximum is 1.0), the better the fitting effect for the model [55].

**3. Results**

*3.1. Forest Color Palette*

The main colors in per forest color image were extracted with primary k-means clustering (Figure A2). The number of main colors was concentrated in three to five colors (Table A2). Among these, the highest number of images had three main colors, accounting for 62.47% of the total forest color images, followed by the number of images with four main colors, accounting for 22.94%, and the number of images with five main colors was the least, accounting for 16.33%.

We determined thresholds for each hue through secondary k-means clustering based on the H-value of the main colors and visualized the forest colors according to the definitions in the forest color palette. During the construction of the forest color palette, the critical value for the color classification threshold was selected to fill each interval, but actually, each block represented a type of color range, including multiple colors within the threshold value [20]. As a consequence, we quantized the forest color palette into eight color families with 72 colors, including orange [0, 31] ∪ (334, 360], yellow (31, 60], yellow-green (60, 85], green (85, 112], blue-green (112, 156], blue (156, 204], purple (204, 248], and red (248, 334] (Figure 2).

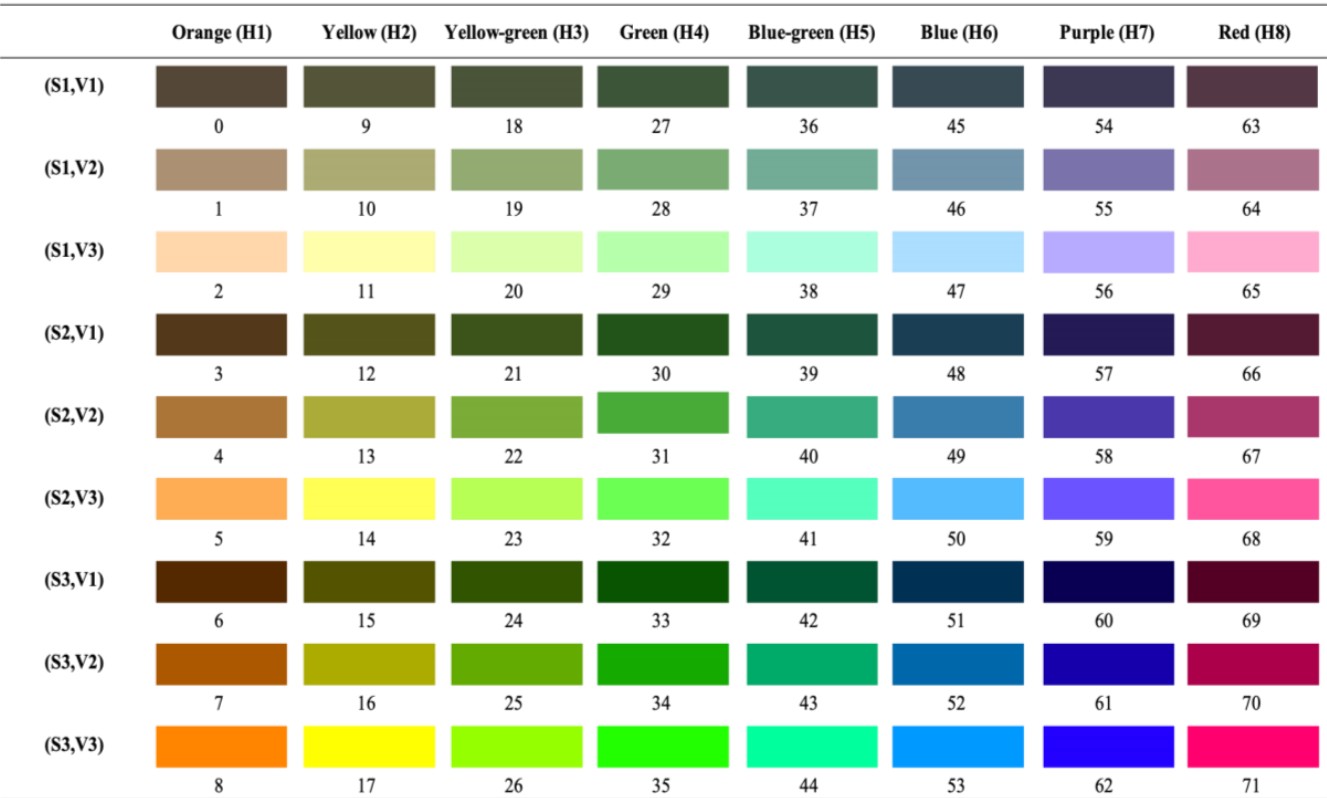

**Figure 2.** Forest color palette which was quantized into eight color families with 72 colors.

### 3.2. Forest Color Patch Characteristics

According to the constructed forest color palette, 470,292 different color patches were interpreted from 20 interior forest landscape images and 20 distant forest landscape images. The results demonstrated that the forest colors were mainly distributed between orange and blue-green (H1–H5). The colors contained in each image are shown in Tables A3 and A4.

The Kruskal–Wallis test results revealed that there were highly significant differences in each color patch index among the different colors and color families for both the interior and distant forest landscapes (Table A6, Figure 3). The color patch indices with a large degree of variation among the color families involved the LPI, C, DIV, COH, and SHDI. The results showed that the orange and yellow color patches had high LPI and COH values, while the green patches had high C, DIV, and SHDI values. These results illustrate that the green patches were more fragmented than the orange and yellow patches in the interior and distant forest landscapes.

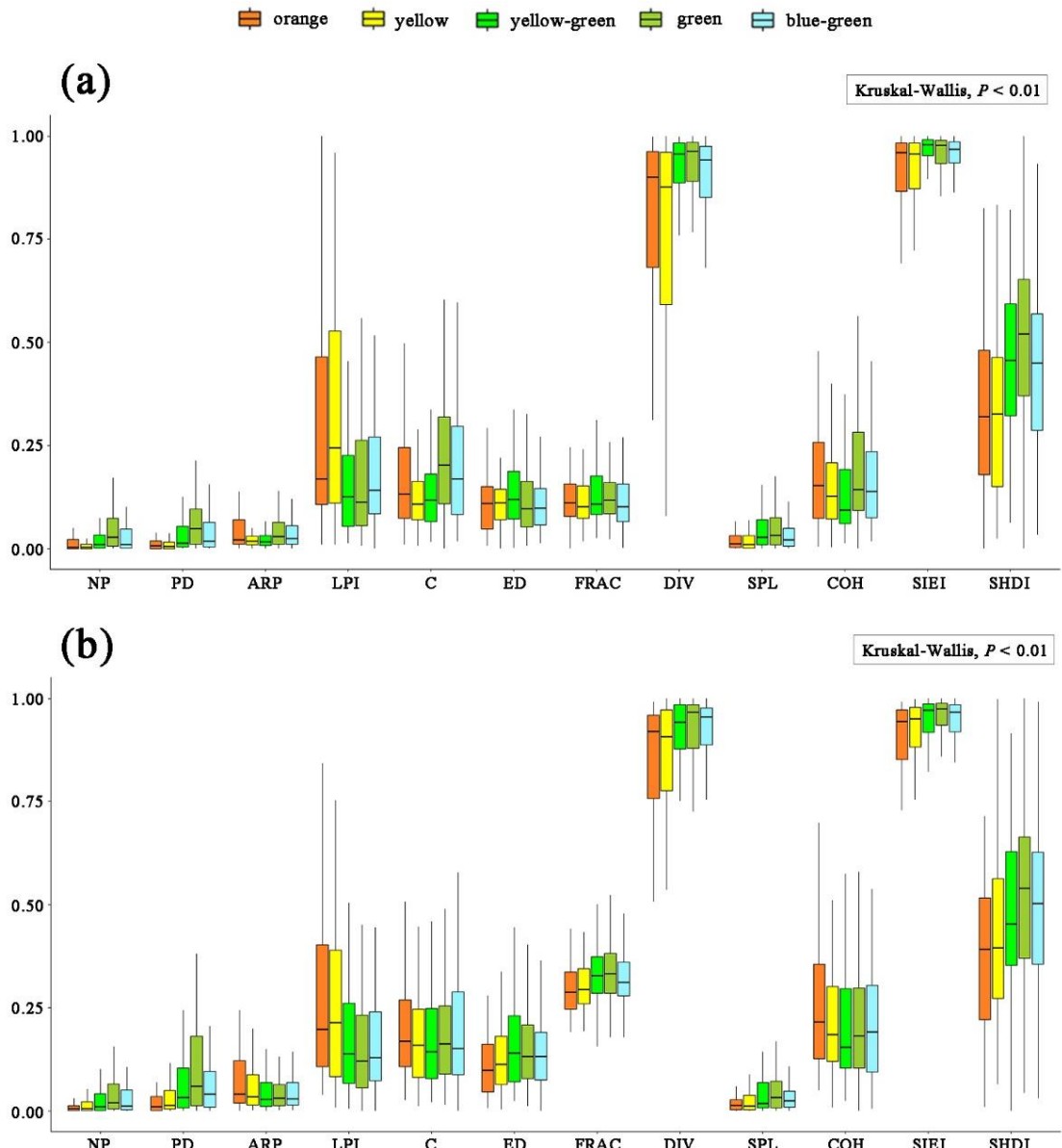

**Figure 3.** (**a**) Color patch indices of interior forest landscape image. (**b**) Color patch indices of distant forest landscape image. Kruskal–Wallis test results for the color patch indices of the different colors. The X-axis is the color patch indices and the Y-axis represents the normalized value of indices. Box plots show the median and quartiles of the color patch indices.

*3.3. Recognition of Forest Color Palette by Humans*

3.3.1. Accuracy of Forest Color Recognition

According to the Chi-square test results (Table A7), there were no significant differences in color recognition accuracy between females and males, nor native and non-native participants in the interior forest landscape ($p > 0.05$). This indicated that the accuracy of the color recognition in the interior forest landscape was not affected by the differences in the participants' gender or place of origin. There was no significant difference in color recognition accuracy in the distant forest landscapes between the different origins of the participants, but there was a significant difference between the different genders ($\chi^2 = 4.355$, $p < 0.05$), with the accuracy of color recognition being higher in males than that in females.

There was a significant difference in the accuracy of color recognition in the interior forest landscape among colors ($\chi^2 = 954.020$, $p < 0.01$). The number of participants who accurately identified colors in descending order was green > orange > yellow-green >

yellow > blue-green (Figure A5). There was a significant difference in the accuracy of color recognition in the distant forest landscape among colors ($\chi^2$ = 1083.400, $p$ < 0.01). The number of participants who accurately identified colors in descending order was green > yellow-green > yellow > orange > blue-green (Figure A6). In conclusion, green had the highest color recognition accuracy, and blue-green had the lowest.

3.3.2. Sensitivity of Forest Color Recognition

According to the Kruskal–Wallis test results (Table A7), there was no significant difference in color recognition sensitivity in the interior forest landscape among the different types of participants (different genders and different places of origin, $p$ > 0.05). There was no significant difference in color recognition sensitivity in the distant forest landscapes between the different origins of participants, and there was a significant difference between the different genders ($\chi^2$ = 21.258, $p$ < 0.05), with the accuracy of color recognition being higher in males than that in females. There was no significant difference ($p$ > 0.05) in color recognition sensitivity among the colors in either the interior or distant forest landscapes.

*3.4. Effect of Forest Color Patch Indices on Human Color Recognition Accuracy*

The regression analysis showed that the color patch index fit the image color recognition accuracy well, but the model fit the image color recognition accuracy better (McFadden's Pseudo $R^2$(i) = 0.578, McFadden's Pseudo $R^2$(d) = 0.433).

For interior forest landscapes, the ARP and FRAC had significant positive effects on the color recognition accuracy, i.e., the larger the color patch area and the more complex the shape, the higher the color recognition accuracy. The NP and PD had a significant negative impact on the color recognition accuracy, i.e., the more color patches, the lower the accuracy of the color recognition (Table 2).

**Table 2.** Results of the logistic regression model for color recognition accuracy. The abbreviations of the various indices are expanded in Table 1.

| Image Type | Index | Exp(coef) | 95% CI | $p$ |
|---|---|---|---|---|
| | (Intercept) | 0.07436 | (0.032, 0.175) | 0.000 ** |
| | NP | 0.99963 | (0.999, 1.000) | 0.000 ** |
| | PD | 0.99954 | (0.999, 1.000) | 0.000 ** |
| | ARP | 1.00047 | (1.000, 1.001) | 0.037 * |
| | DIV | - | - | - |
| | C | 0.99894 | (0.996, 1.002) | 0.426 |
| Interior forest landscape | ED | 0.91135 | (0.815, 1.021) | 0.106 |
| | FRAC | 1.63493 | (1.142, 2.303) | 0.006 ** |
| | SPL | 1.00086 | (1.000, 1.002) | 0.15 |
| | COH | - | - | - |
| | SIEI | 0.57600 | (0.282, 1.192) | 0.133 |
| | SHDI | 1.07155 | (0.992, 1.159) | 0.082 |
| | McFadden's Pseudo $R^2$(i) = 0.578 | | | |
| | (Intercept) | 1.74127 | (0.575, 5.205) | 0.323 |
| | NP | 0.99978 | (1.000, 1.000) | 0.000 ** |
| | PD | 1.00042 | (1.000, 1.001) | 0.000 ** |
| | ARP | 0.99987 | (1.000, 1.000) | 0.473 |
| | DIV | 0.35019 | (0.171, 0.722) | 0.004 ** |
| | C | 1.00177 | (1.000, 1.004) | 0.097 |
| Distant forest landscape | ED | 0.83288 | (0.698, 0.992) | 0.042 * |
| | FRAC | 0.94441 | (0.577, 1.540) | 0.819 |
| | SPL | 1.00004 | (0.999, 1.001) | 0.913 |
| | COH | 0.96898 | (0.952, 0.986) | 0.000 ** |
| | SIEI | - | - | - |
| | SHDI | 1.09751 | (1.009, 1.194) | 0.031 * |
| | McFadden's Pseudo $R^2$(d) = 0.433 | | | |

Significance: * $p$ < 0.05; ** $p$ < 0.01. No significant variables are not shown.

For distant forest landscapes, both the PD and SHDI had significant positive effects on the color recognition accuracy, i.e., when more patches of different colors were evenly distributed, and more patches of the same color were in the unit area, there was higher accuracy in color recognition. The NP, ED, DIV, and COH had significant negative effects on the color recognition accuracy, i.e., the higher the degree of fragmentation for the color patches and the more significant the edge effect, the lower the accuracy of the color recognition (Table 2).

## 4. Discussion

### 4.1. Forest Color Palette Composition

China is predominantly characterized as being in temperate, warm temperate, or subtropical zones with distinctive seasonal climatic features. The vegetation is mainly deciduous broad-leaved forests and evergreen broad-leaved forests, with rich and diverse forest colors and distinct landscape changes [56]. Through quantitative analysis of the real-world forest color composition, a unified forest color palette was produced, which facilitates the horizontal comparison of forest color research results.

The k-means clustering method was originally proposed for pattern recognition problems (for computational classification of a dataset into corresponding categories based on sample characteristics) [24,57,58] and is a relatively common unsupervised algorithm that is considered advantageous due to its good clustering ability and simple operation [25]. The k-means clustering method was utilized in this investigation for the construction of a forest color palette. Previous studies have shown that hue is less influenced by external factors such as lighting and viewing distance when compared to saturation and value [59]. Hue is also the main basis for distinguishing different colors [4]. Therefore, the H-value threshold delineation was used as the focus of the forest color palette constructed in this study. The method described by Xu et al., (2019) [60] was used to obtain the theme colors for each forest color image using the first clustering results, and a second clustering using the hue values was used to obtain the final eight classes of the primary hues.

To the best of our knowledge, this study is the first to attempt to construct a forest color palette using the k-means clustering method. The results were classified into 72 colors with the following eight color families: orange, yellow, yellow-green, green, blue-green, blue, purple, and red. Compared with the methodology of previous studies [61–63], cyan and magenta, which appear less frequently in the forest landscape, were simplified and subsumed in this study, and the forest base color (green) was classified in detail as yellow-green, green, and blue-green. The color composition of the 40 color-matched images was analyzed using the forest color palette constructed in this study. The forest colors were mainly distributed in the H1–H5 range (orange to blue-green). This validated the findings of previous studies [16,20], indicating that there are fewer plants in the red, blue, and purple categories in the natural environment.

### 4.2. Accuracy of Forest Color Recognition by the Human Eyes

Color recognition accuracy was not affected by participant gender for the interior forest landscapes; however, it was influenced by gender for the distant forest landscapes. The results of the Chi-square test were different between the interior and distant forest landscapes, possibly because the color recognition effects of the human eyes were influenced by the observation distance [64]. It was previously reported that as the observation distance increases, the field of view of the forest landscape increases, and thus the ability of the human eye to discriminate chromatic aberrations is improved accordingly [65]. There was no significant difference in color recognition accuracy in the interior forest landscape between genders, similar to the findings of Jiang et al., (1987) [66]. This may be related to the fact that the human visual system usually maintains a highly stable perceptual experience of color [67]. However, there were significant differences in color recognition accuracy between genders in the distant forest landscape, and the recognition accuracy of males was higher than that of females. This may be because the recognition accuracy of

different genders is related to their color preferences [68]. The forest landscape base color (green) is probably more preferred by males [68] and, therefore, more accurate for male color recognition.

There was a significant difference in color recognition accuracy among the different colors, highlighting the fact that human eyes have different abilities to distinguish different colors [69]. Previous studies have shown that the highest level of discrimination by the human eye occurs for green, followed by red, and the lowest level is for blue [30,70]. The results of this investigation agreed, as the highest level of discrimination was found for green [71] and the lowest for blue-green. This could be attributed to the fact that in the human eye, there are 40 times more green retinal receptors in the photoreceptor area when compared with the blue retinal cells [72]. The peak sensitivity of the human eye to light is in the green area [71,73], but only approximately 50% of the light in the blue-green area can reach the retina [70]. However, this may also be related to the environment humans live in, where our color vision system responds to changes in the external environment in a compensatory manner [74]. In ancient times, humans lived in the forest [75], saw more green leaves, and had to pay attention to the changes in the surrounding environment during hunting; there was thus a requirement to be more sensitive to the green colors found in leaves. In future forest color studies, we could therefore consider quantifying more finely for the green areas and less for the blue-green areas.

### 4.3. Sensitivity of Forest Color Recognition by the Human Eyes

Color recognition sensitivity was not affected by participant gender in the interior forest landscapes; however, it was influenced by gender in the distant forest landscapes. The results of the Kruskal–Wallis test were different between the interior and distant forest landscapes, which may be related to the characteristics of the images themselves. The average area for the color patches in the distant forest landscape was larger, and the color contrast of the neighboring patches was more intense, thus reducing color discrimination difficulties by the human eye. The higher color recognition sensitivity of the males, when compared with that of the females for the distant forest landscapes, indicated that males took a shorter time in the forest color matching process, which may be related to the different personalities between males and females, with males being more decisive in their choices compared to females [68,76,77]. In addition, males may be more responsive than females [78,79], being able to quickly click on the colors they see. The close viewing distance in the interior forest landscape makes it easy to be influenced by other factors such as plant texture, form, and size, thus increasing the difficulty for color identification, which may lead to difficulty in reflecting the advantages of male character and responsiveness in color matching.

There was no significant difference in color recognition sensitivity among colors, indicating that the color matching time for the different colors was similar. This may be because color recognition sensitivity is more influenced by human factors, such as the participants' own perceptual ability and responsiveness and less related to the physical properties of the color itself.

### 4.4. Effect of the Color Patch Indices on Human Color Recognition Accuracy

The results showed that the forest color patch index could explain 57.8% and 43.3% of the recognition accuracy in the interior and distant forest landscapes, respectively. The model interpretation degree was good, indicating that the color patch index played an important role in forest color recognition accuracy.

The factors influencing color recognition accuracy varied between the interior and distant forest images, and in general, the number and area of the different color patches were considered the main influencing factors. Cao et al., (2021) [15] also showed that the patch index, which is strongly influenced by color classification and viewing distance, includes characteristics in terms of number, area, diversity, and edges of the color patches. The number and area of the forest color patches are mainly influenced by the spatial layout

of the plants. The higher fragmentation of the plant patches for the same species or similar color possibly leads to an increase in the number of color patches and a decrease in their average area. Hu et al., (2010) [80] found that the average area of urban park landscape patches was positively correlated with the proportion of native tree species, i.e., as the proportion of native tree species planted increased, so did the average area index of the landscape patches. This study found that the smaller the number of color patches and the larger the average area, the higher the accuracy of color recognition. This is probably because the overall difficulty in color matching decreases when the number of color patches is less and the area is larger.

The color recognition accuracy in the interior forest landscape was also influenced by the shape of the color patches. Previous studies have found that the human eye is more sensitive to color changes in smooth areas [19]. However, our study found that the more complex the shape, the higher the human color recognition accuracy was. This may be because complex shapes are more likely to attract human attention in the color matching process [81], thereby resulting in better recognition. Li et al., (2018) [82] showed that the landscape shape index was influenced by a combination of stand density, irrigation and grass cover, and tree trunk morphology. The complexity of the color patch shape was reduced by increasing the stand density and irrigation cover and increased by curved tree trunks when the stand density was at a medium to a high level. Therefore, we can make reasonable plant selection and configuration to increase the complexity of color patch shapes in small-scale forest color landscapes, which enhances the accuracy of human eye color recognition and improves interest in forest landscapes [83].

The color identification accuracy of the distant forest landscapes was also influenced by the color patch index in terms of both dispersion and diversity. The greater the color patch edge density and the more spatially dispersed may cause deeper fragmentation of forest color patches [84]. However, the deepening of color patch fragmentation will distract people's attention which makes the accuracy of human eye color recognition decrease [85]. Li et al., (2021) [86] showed that the degree of landscape patch fragmentation was significantly and negatively correlated with the diversity of shrubs. This means that the connectivity among forest color patches can be increased by adding shrub diversity, leading to reduced fragmentation in color landscape patches. On the other hand, the greater the color patch diversity index, i.e., the more balanced the color distribution, the higher the color recognition accuracy. A balanced mix of primary, secondary, and accent colors presents a harmonious and unified visual effect, providing a quiet and stable psychological feeling, which may make participants more focused on color matching and help to improve their color recognition accuracy [87]. Therefore, it is necessary to pay attention to the diversity and balance of color matching in the creation of larger-scale forest color landscapes [20].

*4.5. Limitations and Research Prospects*

In terms of image sources, forest images were obtained from the Sina Weibo platform, which has the advantage of a large sample size but has problems such as inconsistent image quality. On the one hand, image quality is affected by human factors such as the equipment and mode used by the photographer. On the other hand, photos uploaded to the Sina Weibo platform and the final forest images selected are inevitably influenced by the preferences of the photographer and the researcher. As a result, there may be deviations between color information and actual forest color [88]. In the future, the same equipment and modes of photographing can be used to ensure image quality. Adopting algorithms for automatic image selection can reduce the influence of personal preferences on study results. In addition, this study focused only on color as the most important visual element in people's perception of forest landscapes. In the future, we can consider the role of other visual factors (such as size and shape) and non-visual factors (such as smell and sound) on forest landscape perception in order to guide forest landscape creation and enhancement more precisely.

## 5. Conclusions

In this study, a forest color palette was constructed by quantifying the color information in forest images into 72 colors and eight color families. Color matching experiments were conducted to assess the accuracy and sensitivity of human eyes for forest color recognition. It was determined that color discrimination accuracy for humans was highest for green and lowest for blue-green. Based on the constructed forest color palette, color patch indices were calculated for 40 color-matched images, and the effects of the interior and distant forest color indices on human color recognition were explored, respectively. We found that a smaller number of color patches, a larger average area, and a more complex shape would lead to an increase in color identification accuracy for interior forest landscapes. A more balanced distribution of the different colored patches and a lower degree of fragmentation would result in more accurate color recognition for distant forest landscapes. Consequently, we suggest increasing the area of the target color appropriately and reducing its fragmentation when creating forest color landscapes. For small-scale forest color landscapes, increase the complexity of color patch shapes to enhance landscape interest. For larger-scale forest landscapes, attention should be paid to the balance of color matching. Overall, this study has improved our knowledge of the forest color palette and explored the effects of the color patch indices on the accuracy and sensitivity of forest color recognition from the perspective of human eye perception, providing new insights for forest color quantification and forest landscape planning research, which is conducive to creating forest landscapes that better meet human visual aesthetic needs. However, there were also limitations to this study, and to address these in the future, we aim to improve picture collection, picture processing, and color extraction, and carry out more extensive research, to further the available references for the planning and construction of forest color landscapes.

In the process of creating the forest color landscape, we suggest appropriately increasing the proportion of native tree species and paying attention to the layout of plants. Moreover, we suggest that plants with similar colors should be planted centrally, and on this basis, an appropriate amount of contrast plants should be planted for embellishment [89], which will attract people's attention. In terms of small-scale forest color landscape design, we suggest appropriately reducing stand density and the coverage of irrigation and grass. It may even be possible to create a solitary landscape by using tree species with graceful shapes, exotic postures, unique colors, and high ornamental value, increasing the color shape complexity of the plant itself and giving people sufficient space for their imagination. In terms of larger-scale forest color landscape design, we suggest that the diversity of plant species should be appropriately increased, and shrubs should be reasonably matched with the trees to increase the hierarchy of the forest color landscape.

**Author Contributions:** Conceptualization, C.W. and W.H.; methodology, W.H. and C.Z.; software, W.H. and C.Z.; validation, C.Z., C.W. and W.H.; formal analysis, C.Z., C.W. and W.H.; resources, C.W. and C.Z.; data curation, W.H.; writing—original draft, W.H.; writing—review & editing, C.Z., C.W., L.Y. and W.H.; visualization, W.H.; supervision, C.Z. and C.W.; project administration, C.Z. and C.W.; funding acquisition, C.W. and C.Z. All authors have read and agreed to the published version of the manuscript.

**Funding:** This research was funded by National Non-Profit Research Institutions of the Chinese Academy of Forestry (CAFYBB2020ZB008), National Natural Science Foundation of China (No. 31800608).

**Informed Consent Statement:** Informed consent was obtained from all subjects involved in the study.

**Data Availability Statement:** The datasets generated during and/or analyzed during the current study are available from the corresponding author on reasonable request.

**Conflicts of Interest:** The authors declare no conflict of interest.

## Appendix A

**Table A1.** Summary data of the images collected from the different forest parks and used in this investigation.

| Region | No. | National Forest Park | Province | Number of Images | Total |
|---|---|---|---|---|---|
| North China | 1 | Saihanta National Forest Park | Hebei | 30 | |
| | 2 | Yesanpo National Forest Park | Hebei | 30 | |
| | 3 | Arshaan National Forest Park | Inner Mongolia | 50 | |
| | 4 | Hengshan National Forest Park | Shanxi | 30 | 227 |
| | 5 | Taihang Canyon National Forest Park | Shanxi | 40 | |
| | 6 | Labagou Origin Forest Park | Beijing | 30 | |
| | 7 | Jiulongshan National Forest Park | Tianjin | 17 | |
| Northeast China | 1 | Xianglushan National Forest Park | Heilongjiang | 12 | |
| | 2 | Wuying National Forest Park | Heilongjiang | 18 | |
| | 3 | Lafashan National Forest Park | Jilin | 22 | 92 |
| | 4 | Daheishan National Forest Park | Liaoning | 26 | |
| | 5 | Dalian Tianmen Mountain National Forest Park | Liaoning | 14 | |
| East China | 1 | Huangshan National Forest Park | Anhui | 28 | |
| | 2 | Tachuan National Forest Park | Anhui | 30 | |
| | 3 | Fuzhou National Forest Park | Fujian | 30 | |
| | 4 | Yangling National Forest Park | Jiangxi | 30 | |
| | 5 | Taishan National Forest Park | Shandong | 26 | 224 |
| | 6 | Yandang Mountain National Forest Park | Zhejiang | 16 | |
| | 7 | Sheshan National Forest Park | Shanghai | 24 | |
| | 8 | Zijinshan National Forest Park | Jiangsu | 40 | |
| Central China | 1 | Baiyunshan National Forest Park | Henan | 11 | |
| | 2 | Shennongjia National Forest Park | Hubei | 30 | 68 |
| | 3 | Zhangjiajie Tianmen Mountain National Forest Park | Hunan | 27 | |
| Northwest China | 1 | Tulugou National Forest Park | Gansu | 18 | |
| | 2 | Maiji National Forest Park | Gansu | 30 | |
| | 3 | Taibaishan National Forest Park | Shaanxi | 30 | |
| | 4 | Jinsixia National Forest Park | Shaanxi | 30 | 165 |
| | 5 | Tianshan Grand Canyon National Forest Park | Xinjiang | 30 | |
| | 6 | Beishan National Forest Park | Qinghai | 27 | |
| Southwest China | 1 | Fenghuangshan National Forest Park | Guizhou | 15 | |
| | 2 | Leigongshan National Forest Park | Guizhou | 10 | |
| | 3 | Tiantaishan National Forest Park | Sichuan | 20 | |
| | 4 | Xishuangbanna National Forest Park | Yunnan | 30 | 101 |
| | 5 | Geleshan National Forest Park | Chongqing | 18 | |
| | 6 | Segyi La National Forest Park | Tibet | 8 | |
| South China | 1 | Guanyinshan National Forest Park | Guangdong | 26 | |
| | 2 | Wutongshan National Forest Park | Guangdong | 30 | |
| | 3 | Darongshan National Forest Park | Guangxi | 14 | 109 |
| | 4 | Debao Red Leaves National Forest Park | Guangxi | 19 | |
| | 5 | Jianfengling National Forest Park | Hainan | 20 | |
| | | Total | | 986 | |

**Table A2.** Number and proportion of forest images from the different regions assessed in this study with optimal K-values.

| Region | K = 3 | | K = 4 | | K = 5 | | Total |
|---|---|---|---|---|---|---|---|
| | Number of Images | Proportion% | Number of Images | Proportion% | Number of Images | Proportion% | |
| North China | 154 | 67.84 | 45 | 19.82 | 28 | 12.33 | 227 |
| Northeast China | 63 | 68.48 | 16 | 17.39 | 13 | 14.13 | 92 |
| East China | 143 | 63.84 | 41 | 18.3 | 40 | 17.86 | 224 |
| Central China | 32 | 47.06 | 21 | 30.88 | 15 | 22.06 | 68 |
| Northwest China | 97 | 58.79 | 39 | 23.64 | 29 | 17.58 | 165 |
| Southwest China | 57 | 56.44 | 22 | 21.78 | 22 | 21.78 | 101 |
| South China | 70 | 64.22 | 25 | 22.94 | 14 | 12.84 | 109 |
| Total | 616 | 62.47 | 209 | 21.2 | 161 | 16.33 | 986 |

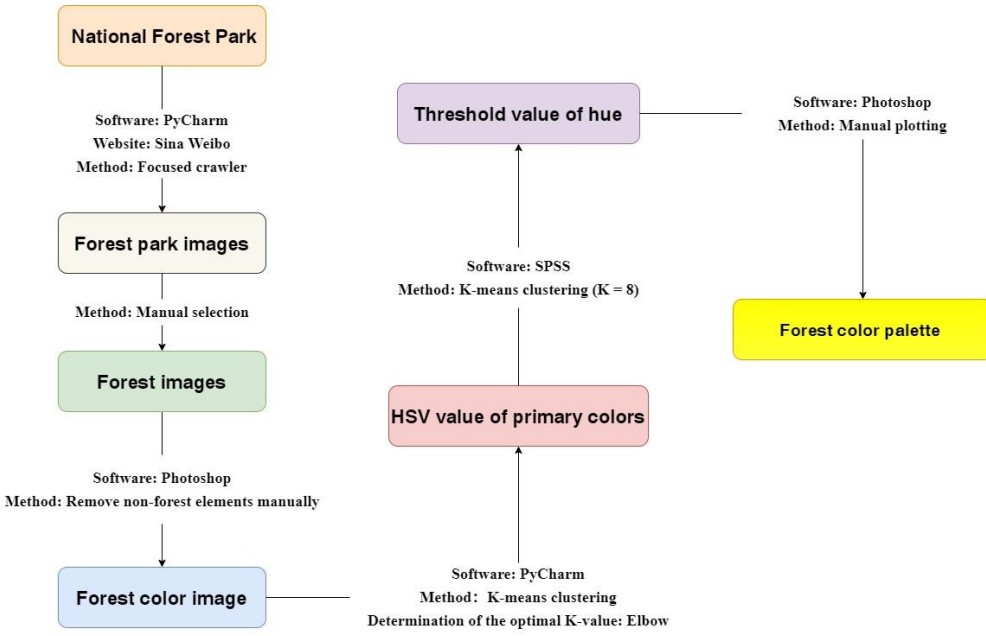

**Figure A1.** Process used to construct a forest color palette.

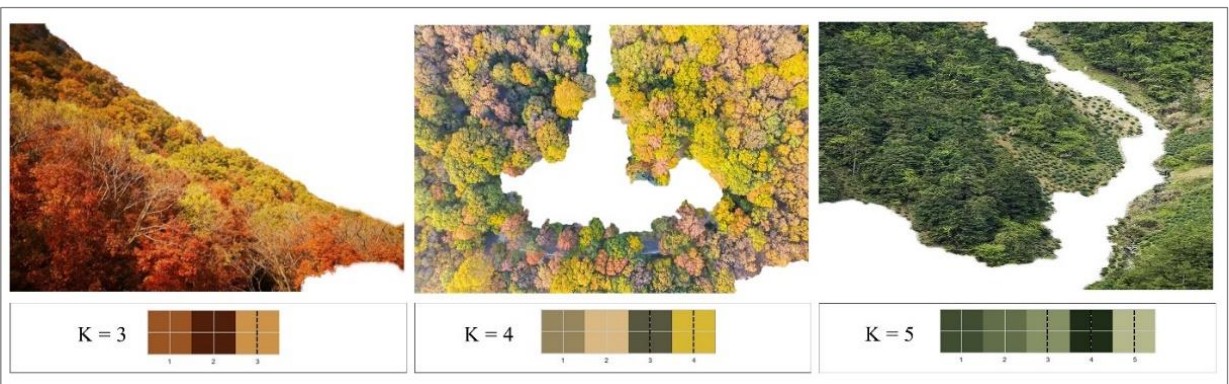

**Figure A2.** The different squares of color represent the theme colors extracted from forest color images by k-means method.

**Table A3.** The numbers and color blocks represent the serial number and composition color of interior forest landscape images respectively.

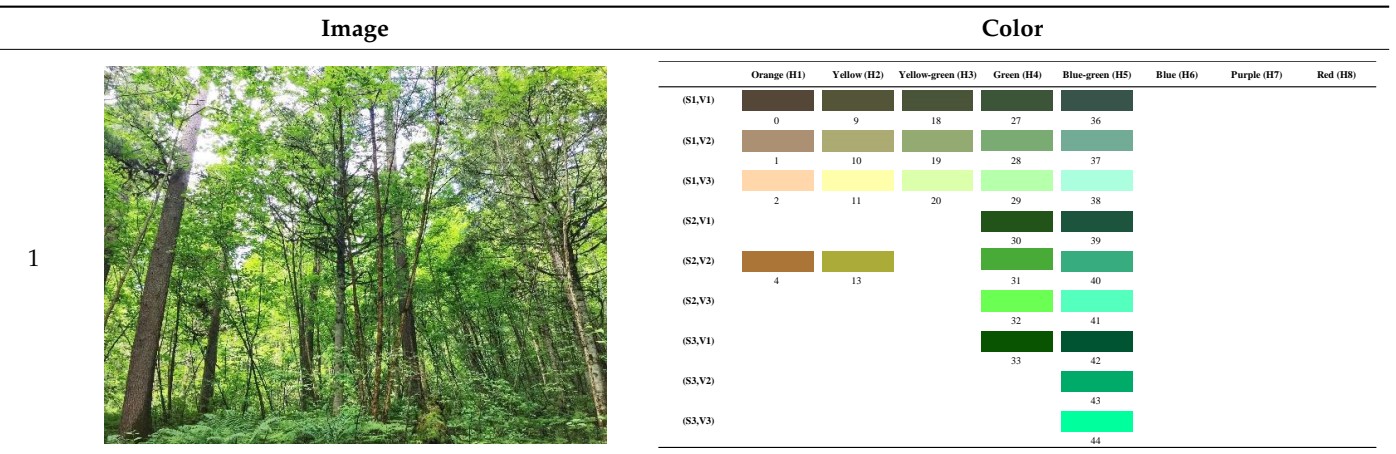

**Table A3.** *Cont.*

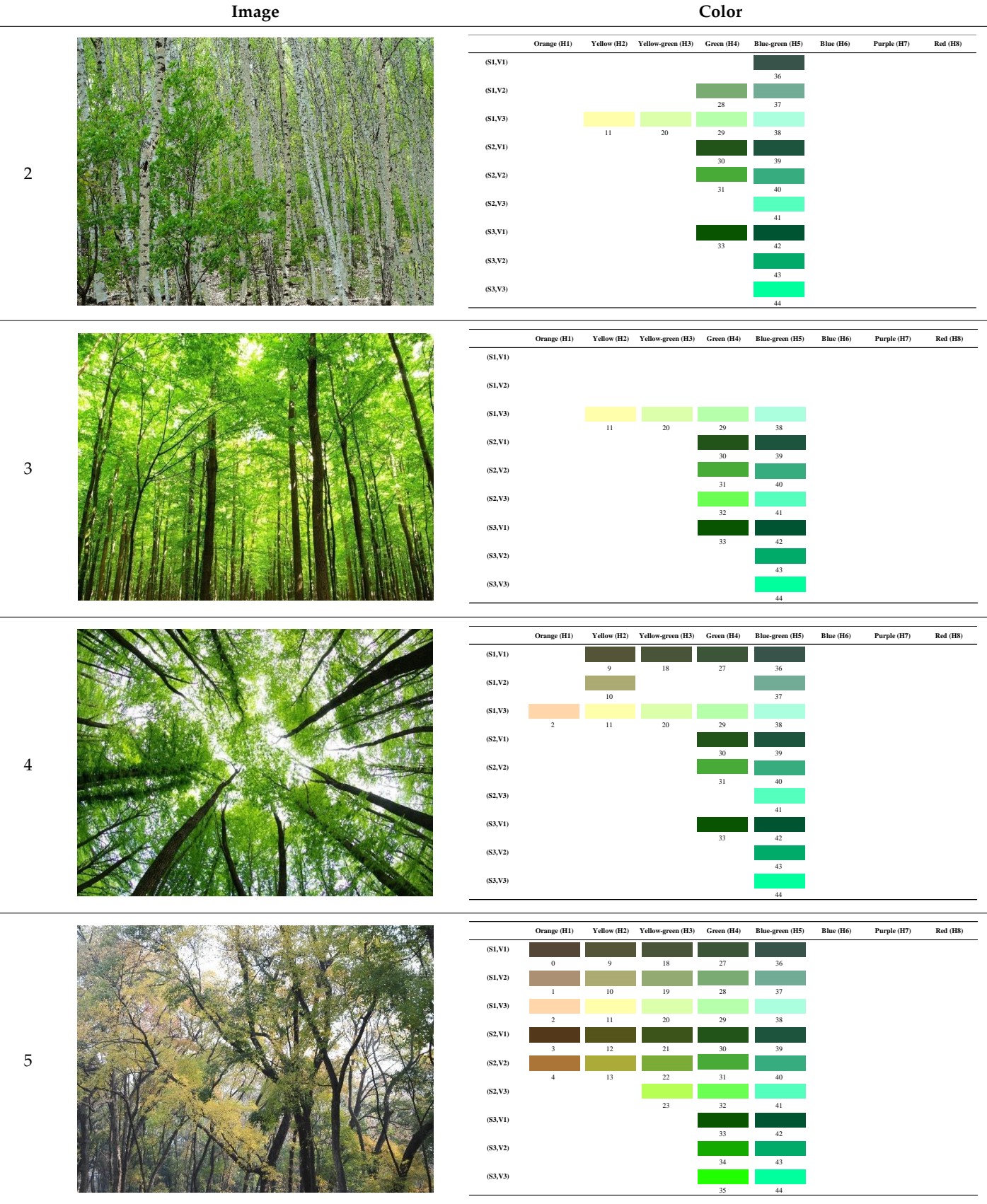

**Table A3.** *Cont.*

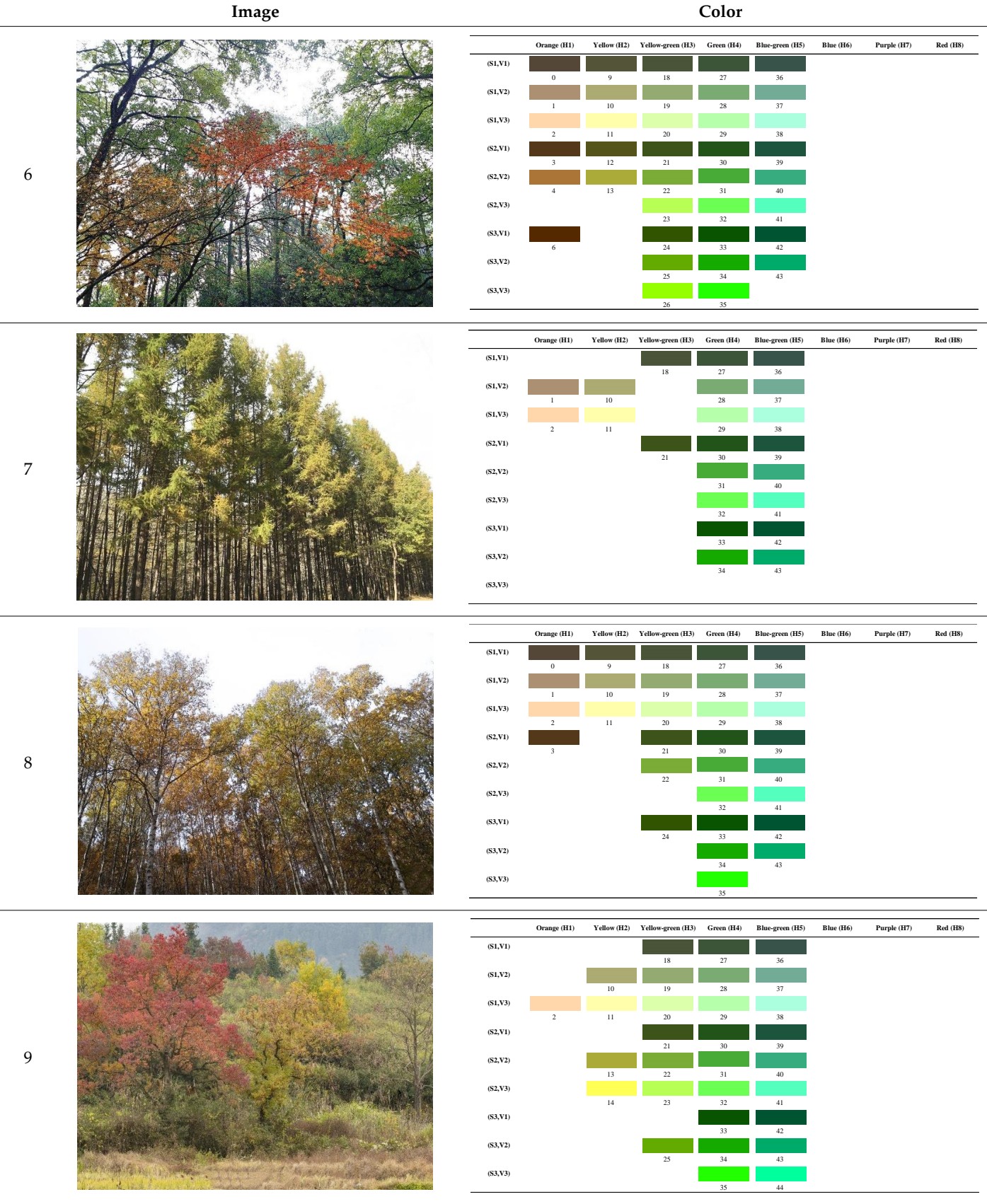

**Table A3.** *Cont.*

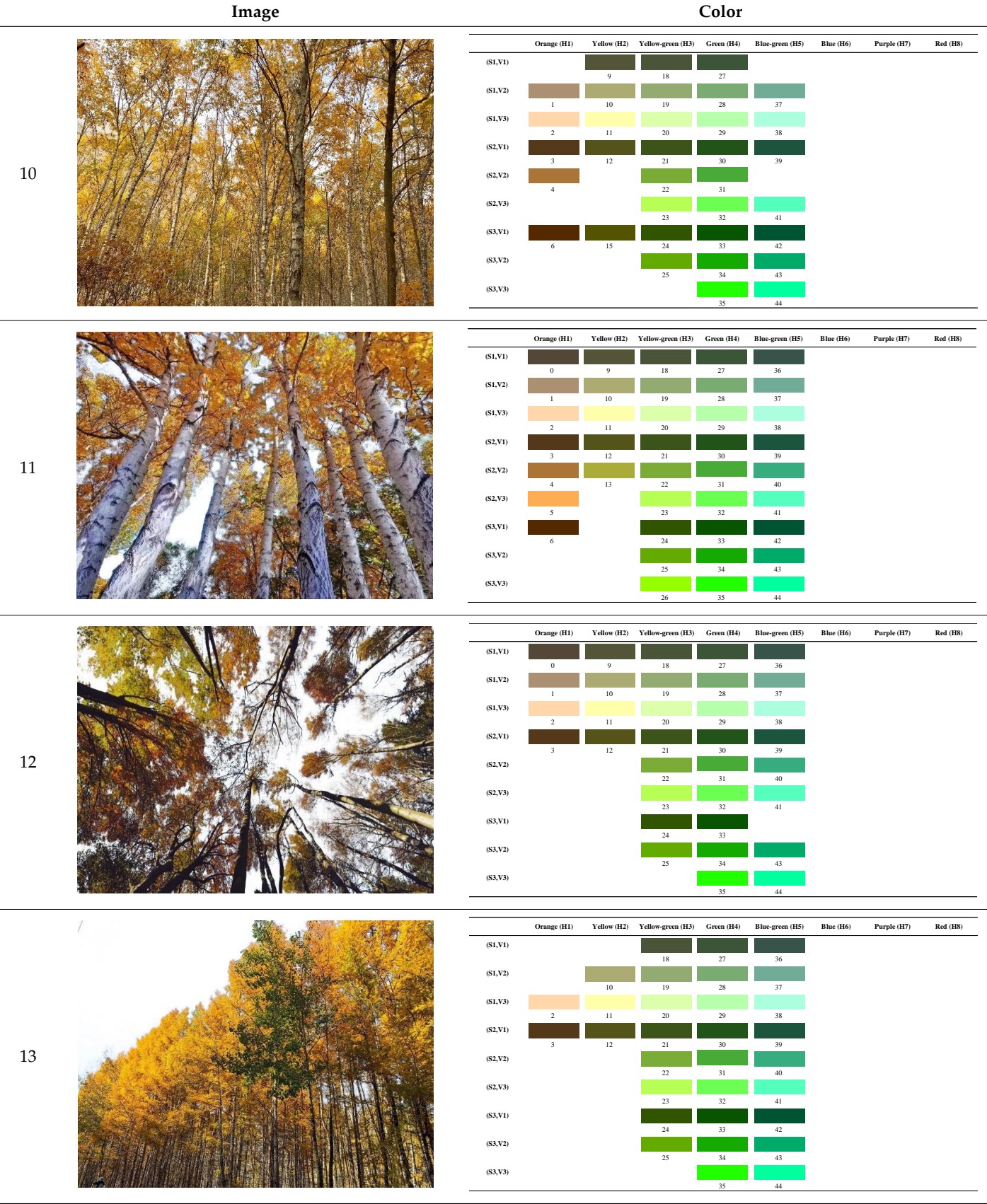

**Table A3.** *Cont.*

| Image | Color |
| --- | --- |
| | |

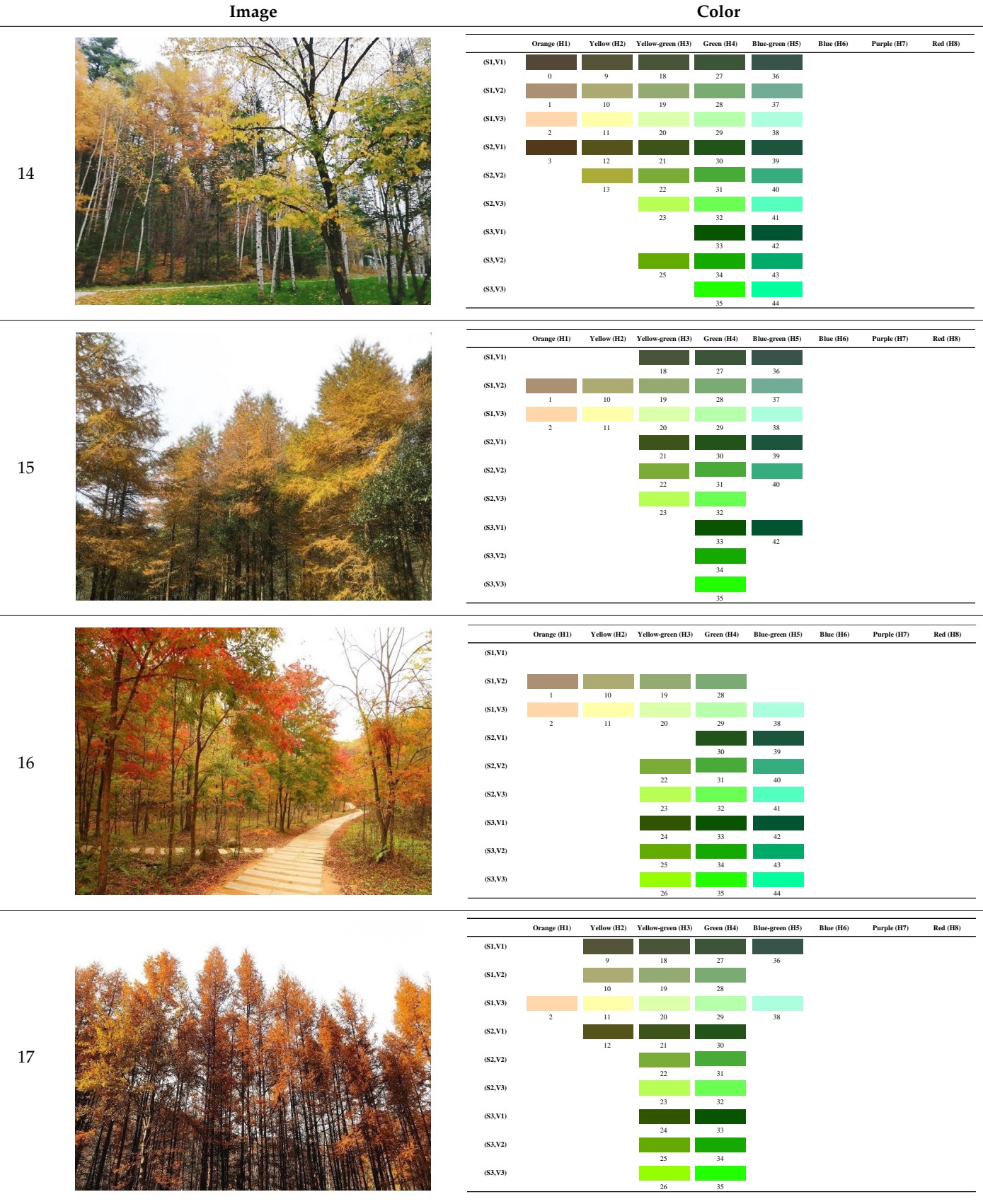

**Table A3.** *Cont.*

| Image | Color |
|-------|-------|

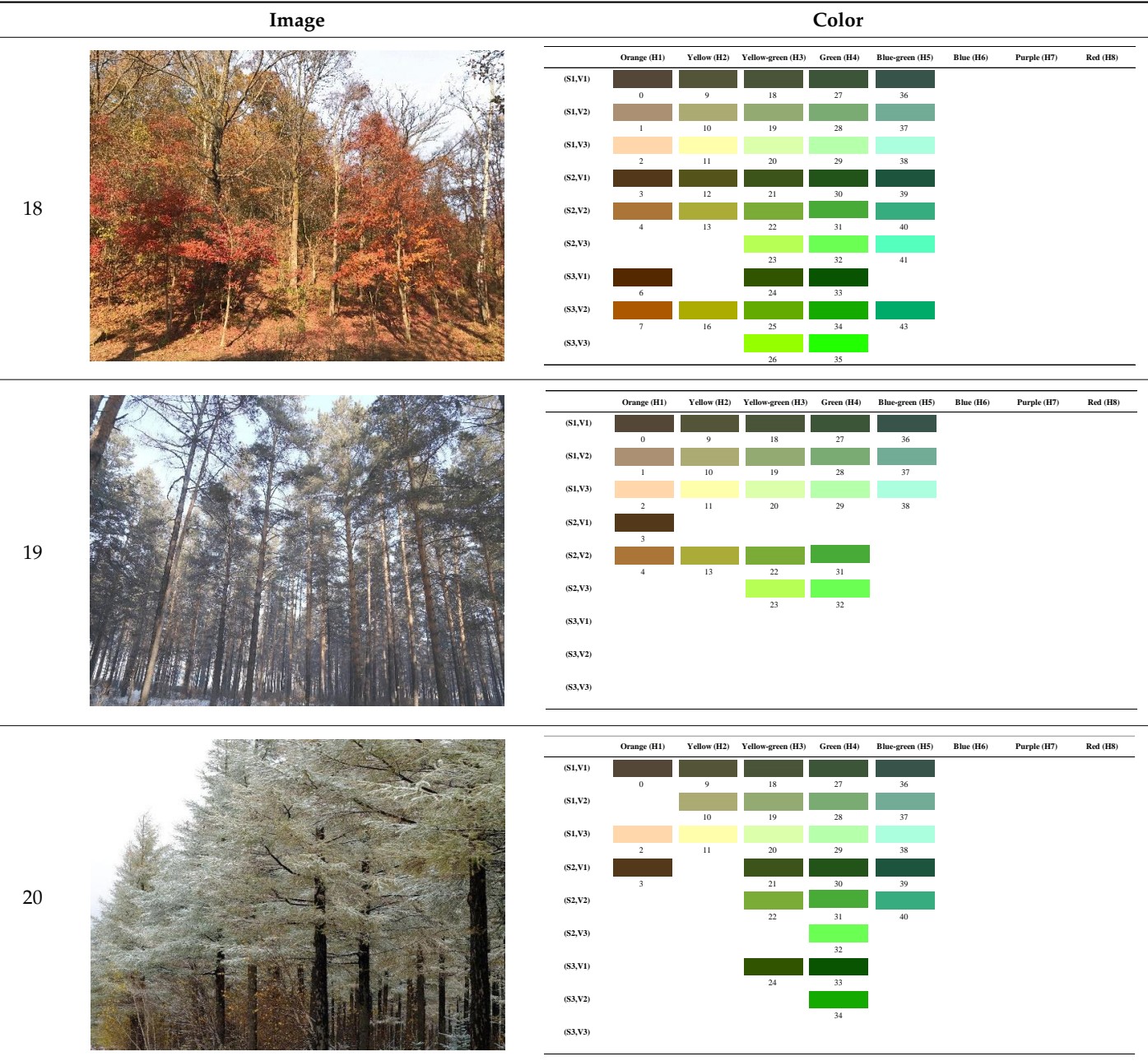

**Table A4.** The numbers and color blocks represent the serial number and composition color of distant forest landscape images respectively.

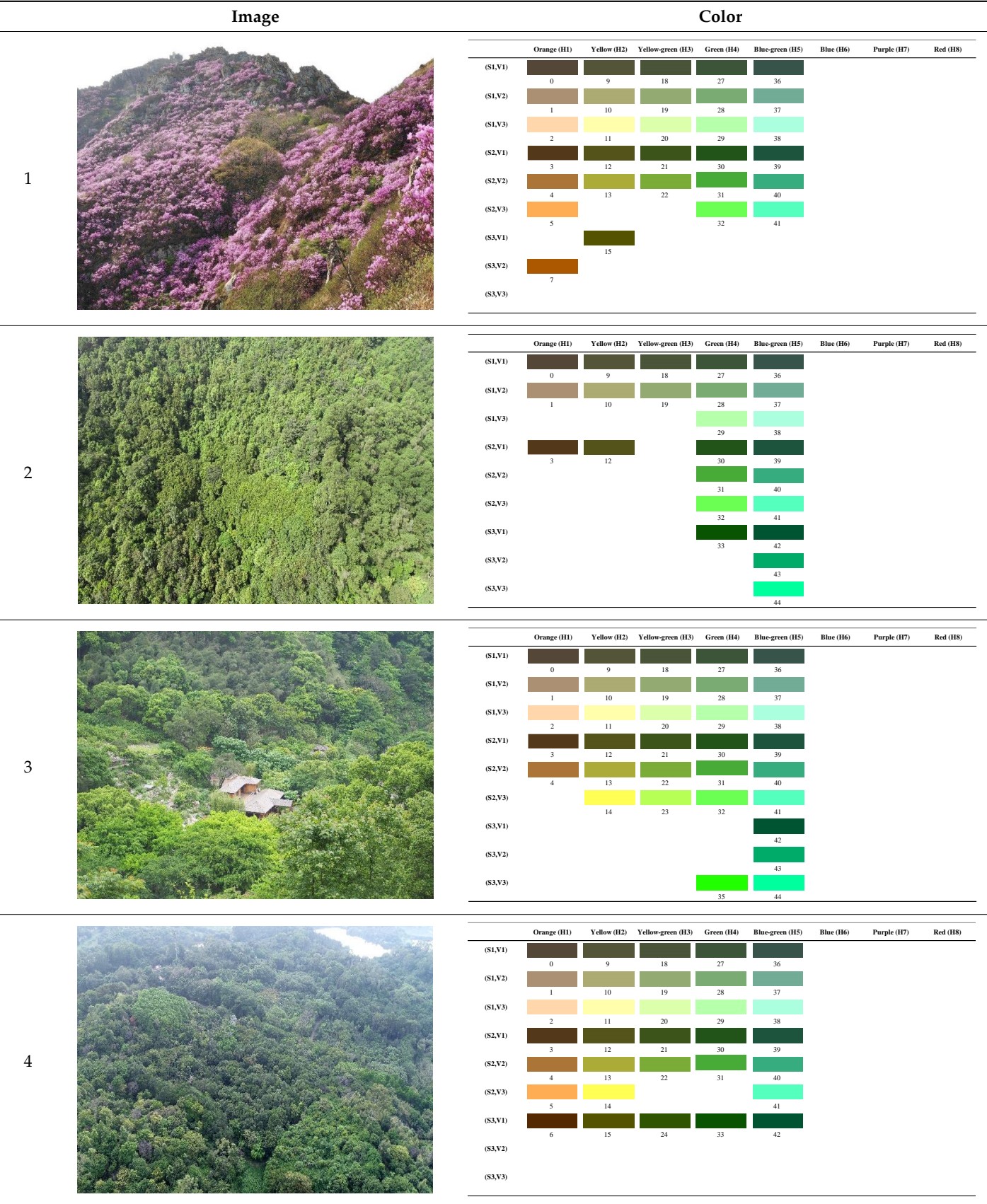

**Table A4.** *Cont.*

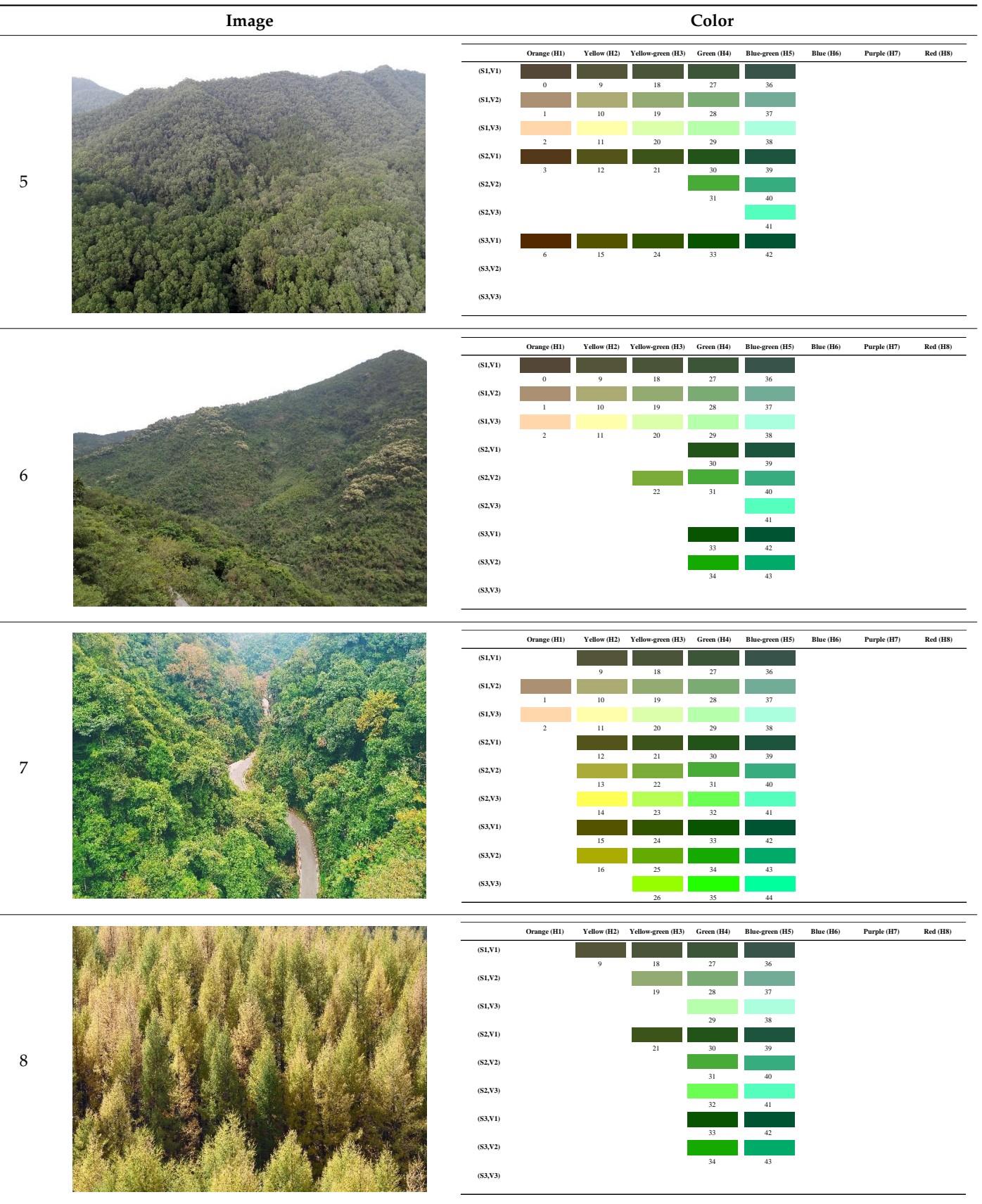

**Table A4.** *Cont.*

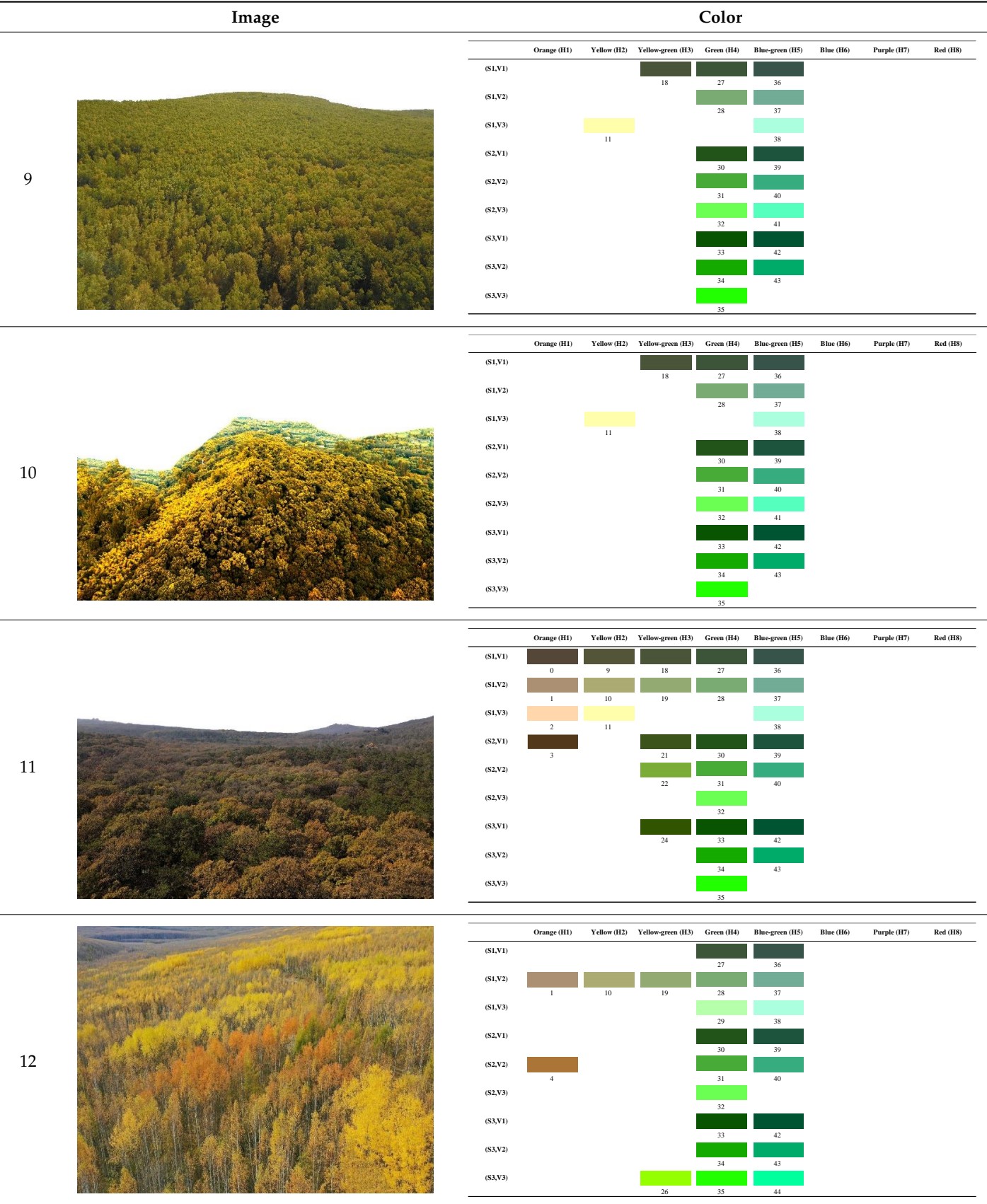

**Table A4.** *Cont.*

| Image | Color |
|---|---|

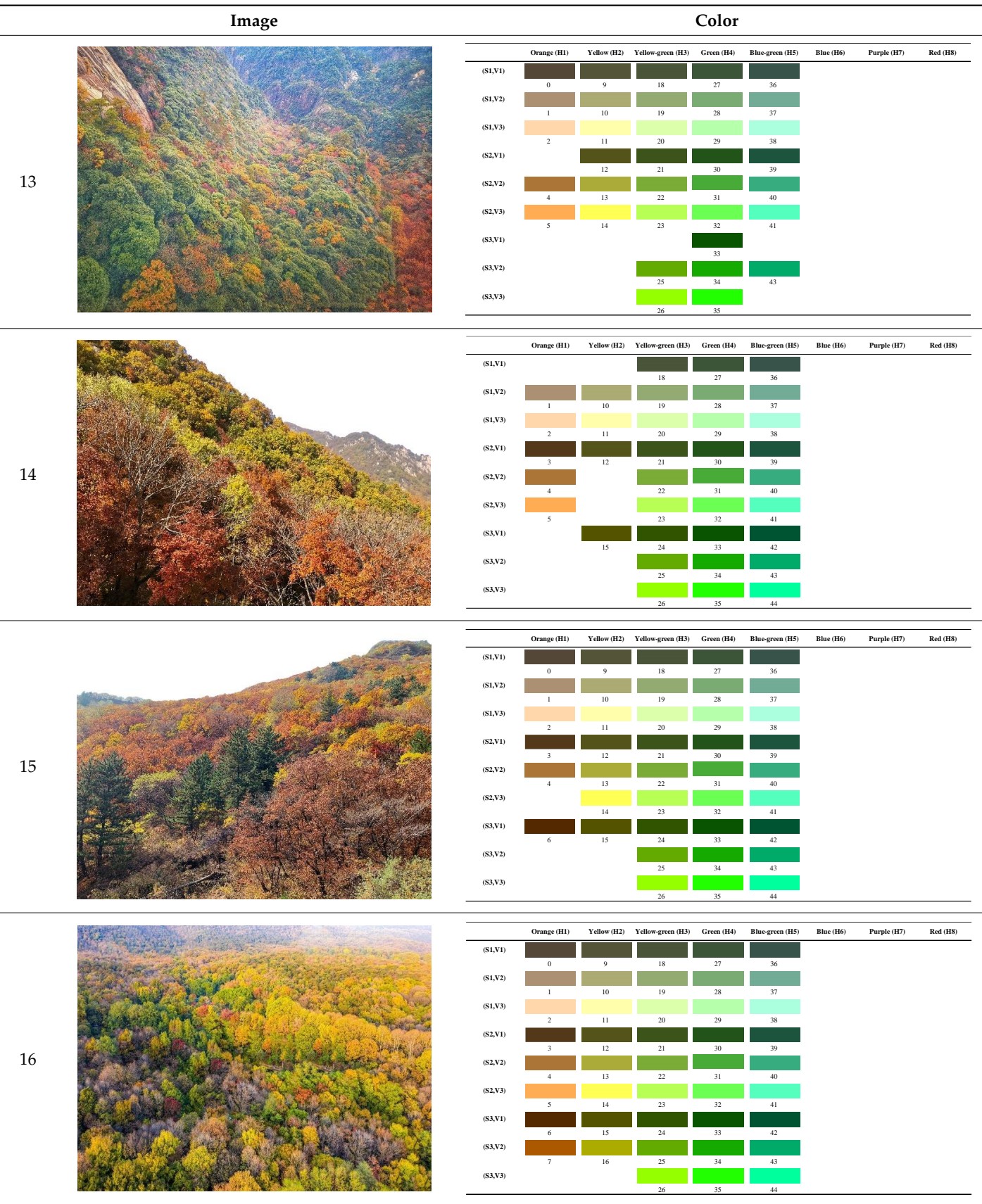

**Table A4.** *Cont.*

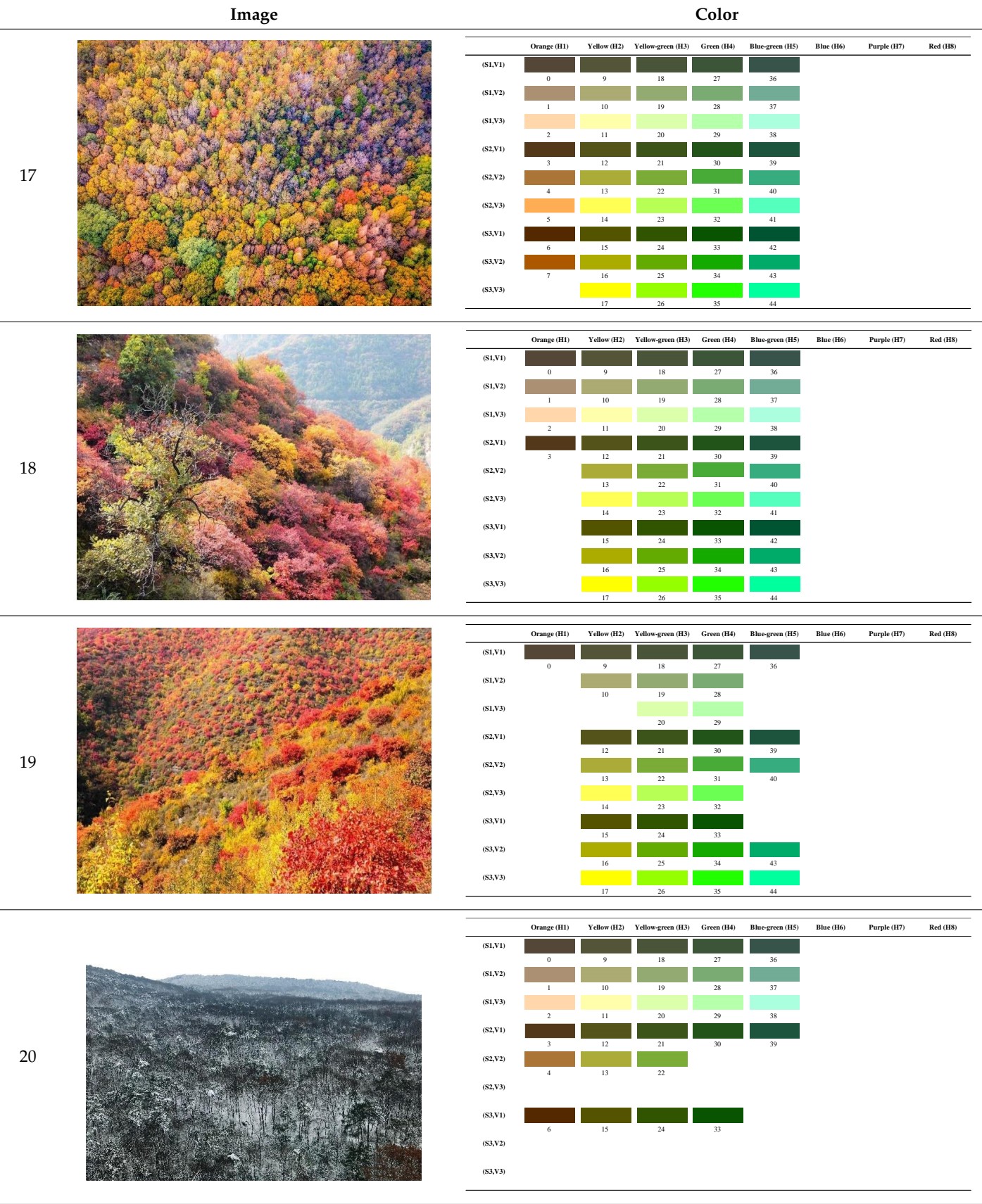

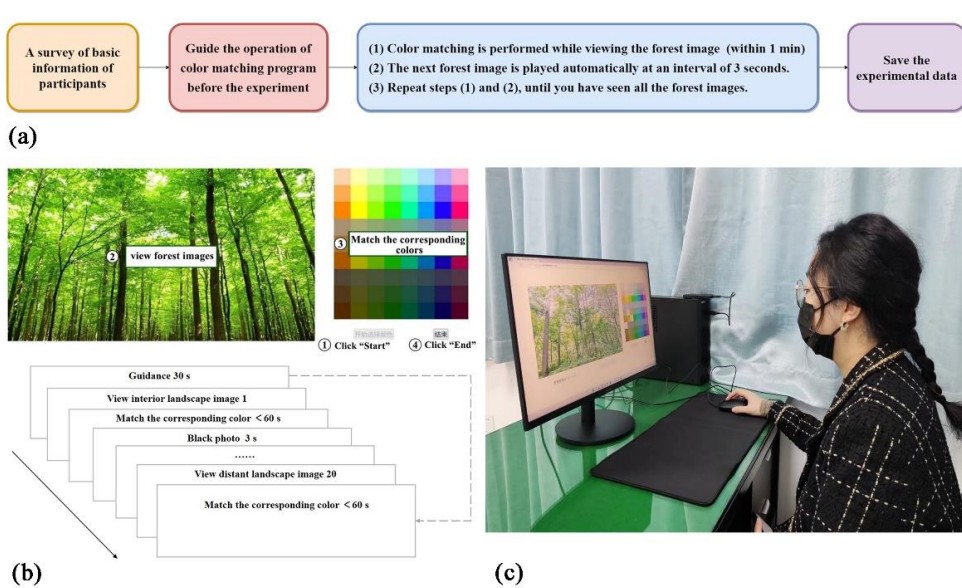

**Figure A3.** (**a**) Experimental procedure used for forest color matching. (**b**) Operating interface of the forest color matching program and the playback sequence of the experimental images. (**c**) Conducting color matching experiment in the lab.

**Table A5.** Calculation formula of color patch indicators.

| No. | Indicators | Formula | Meaning of Parameters |
|---|---|---|---|
| 1 | NP | $NP = N$ | N is the total number of patches |
| 2 | LPI | $LPI = \frac{max(a_{ij})}{A} \times 100\%$ | $a_{ij}$ is the area of patch $ij$, $A$ is the total landscape area |
| 3 | ARP | $ARP = \frac{A}{N}$ | A is the total landscape area, N is the total number of patches |
| 4 | PD | $PD = \frac{n_i}{A} \times 10,000 \times 100$ | $n_i$ is the number of class $i$ patches, $A$ is the total landscape area |
| 5 | C | $C = P_i^*$ | $P_i^*$ is the perimeter of class $i$ patches |
| 6 | ED | $ED = \frac{\sum_{k=1}^m P_i^*}{A}$ | $P_i^*$ is the perimeter of class $i$ patches, $A$ is the total landscape area |
| 7 | FRAC | $FRAC = 2\frac{ln\left(0.25P_{ij}^*\right)}{ln a_{ij}}$ | $P_{ij}^*$ is the perimeter of the patch $ij$, $a_{ij}$ is the area of patch $ij$ |
| 8 | DIV | $DIV = 1 - \sum_{i=1}^m \sum_{j=1}^n \left(\frac{a_{ij}}{A}\right)^2$ | $a_{ij}$ is the area of patch $ij$, $A$ is the total landscape area |
| 9 | COH | $COH = \left[1 - \frac{\sum_{j=1}^m P_{ij}^*}{\sum_{j=1}^n P_{ij}^* \sqrt{a_{ij}}}\right] \times \left[1 - \frac{1}{\sqrt{A}}\right]^{-1} \times 100$ | $P_{ij}^*$ is the perimeter of the patch $ij$, $a_{ij}$ is the area of patch $ij$, $A$ is the total landscape area |
| 10 | SPL | $SPL = \frac{A^2}{\sum_{j=1}^n a_{ij}^2}$ | $a_{ij}$ is the area of patch $ij$, $A$ is the total landscape area |
| 11 | SIEI | $SIEI = \frac{1 - \sum_{i=1}^m P_i^2}{1 - \left(\frac{1}{m}\right)}$ | $P_i$ is the proportion of type $i$ patches, $m$ is the number of patch classes |
| 12 | SHDI | $SHDI = \sum_{i=1}^m (P_i ln P_i)$ | $P_i$ is the proportion of type $i$ patches |

**Table A6.** Results of the Kruskal–Wallis test for the color patch index among the different colors.

| Index | Interior Forest Landscape | | Distant Forest Landscape | |
|---|---|---|---|---|
| | $\chi^2$ | $p$ | $\chi^2$ | $p$ |
| NP | 3482.7 | 0.000 ** | 2629.7 | 0.000 ** |
| PD | 3965.1 | 0.000 ** | 3395.9 | 0.000 ** |
| ARP | 2344.2 | 0.000 ** | 4142.9 | 0.000 ** |
| LPI | 2400.2 | 0.000 ** | 2227.1 | 0.000 ** |
| C | 2741.7 | 0.000 ** | 4056.9 | 0.000 ** |
| ED | 2197.1 | 0.000 ** | 4156.4 | 0.000 ** |
| FRAC | 2101.1 | 0.000 ** | 3318.9 | 0.000 ** |
| DIV | 2711.8 | 0.000 ** | 2312.2 | 0.000 ** |
| SPL | 2711.8 | 0.000 ** | 2312.2 | 0.000 ** |
| COH | 1882.7 | 0.000 ** | 3686.8 | 0.000 ** |
| SIEI | 1919.5 | 0.000 ** | 2700.5 | 0.000 ** |
| SHDI | 3098.1 | 0.000 ** | 2323.6 | 0.000 ** |

Significance: ** $p < 0.01$. No significant variables are not shown.

## Informed Consent Form for Experimental Participants

Please read the following information carefully before you sign to participate in the experiment.

**Protocol Title: Research on the recognition of human eye color based on forest images**

**Principal Investigator:** Wenjing Han, Chang Zhang, Cheng Wang and Luqin Yin

Research Institute of Forestry, Chinese Academy of Forestry, Beijing 100091, China;

Key Laboratory of Tree Breeding and Cultivation and Urban Forest Research Centre, National

Forestry and Grassland Administration, Beijing 100091, China

**Study Contact:** Wenjing Han  **Email:** hanwenjing@caf.ac.cn

| |
|---|
| **Purpose**<br>You have been asked to participate in a research study on color matching of forest images. We would like your permission to enroll you as a participant in this research study. |
| **Procedure**<br>The stimulus images will be presented on a 23.8-inch monitor (screen resolution 1920 x 1080 pixels, 60 Hz). You are required to sit 600-650 mm away from the central monitor. While carefully observing the experimental images, you are required to click on all the colors which you see in the forest color palette according to your color perception. The experiment time is 1 hour. |
| **Confidentiality**<br>The results of this study may be published in an academic journal/book or used for teaching purposes. However, your name or other identifiers will not be used in any publication or teaching materials without your specific permission. |
| **Withdraw from the research**<br>Participation is voluntary, refusal to take part in the study involves no penalty or loss of benefits to which participants are otherwise entitled, and participants may withdraw from the study at any time without penalty or loss of benefits to which they are otherwise entitled. |
| **Experimenter**<br>I have explained the purpose of the research, the study procedures, identifying those that are investigational, the possible discomforts and have answered any questions regarding the study to the best of my ability.<br><br>Signature:\_\_\_\_\_\_\_\_\_\_\_\_\_\_\_\_ Date:\_\_\_\_\_\_\_\_\_\_\_\_\_ |
| **Subject**<br>I confirm that the purpose of the research, the study procedures and possible discomforts that I may experience have been explained to me. All my questions have been satisfactorily answered. I have read this consent form. My signature below indicates my willingness to participate in this study.<br><br>Signature:\_\_\_\_\_\_\_\_\_\_\_\_\_\_\_\_ Date:\_\_\_\_\_\_\_\_\_\_\_\_\_ |

**Figure A4.** Informed consent form for experimental participants.

**Table A7.** Differences in color recognition among the different types of participants.

| Image Type | Participant | Accuracy | | | | Sensitivity | |
|---|---|---|---|---|---|---|---|
| | | 0 | 1 | $\chi^2$ | $p$ | $\chi^2$ | $p$ |
| Interior forest landscape | Gender | | | 0.164 | 0.686 | 0.175 | 0.676 |
| | Male | 7123 | 647 | | | | |
| | Female | 7108 | 662 | | | | |
| | Place | | | 0.001 | 0.976 | 0.011 | 0.917 |
| | Local | 11,187 | 1028 | | | | |
| | Non-local | 3044 | 281 | | | | |
| | Color | | | 954.020 | 0.000 ** | 46.457 | 0.192 |
| Distant forest landscape | Gender | | | 4.355 | 0.037 ** | 21.258 | 0.000 ** |
| | Male | 7718 | 997 | | | | |
| | Female | 7805 | 910 | | | | |
| | Place | | | 0.215 | 0.643 | 0.330 | 0.566 |
| | Local | 12,011 | 1466 | | | | |
| | Non-local | 3512 | 441 | | | | |
| | Color | | | 1083.400 | 0.000 ** | 46.037 | 0.272 |

Significance: ** $p < 0.01$. No significant variables are not shown.

| Color | Accuracy | | Total | Color | Accuracy | | Total |
|---|---|---|---|---|---|---|---|
| | **0** | **1** | | | **0** | **1** | |
| **orange** | **1416** | **324** | **1740** | 25 | 243 | 27 | 270 |
| 0 | 211 | 59 | 270 | 26 | 146 | 4 | 150 |
| 1 | 309 | 51 | 360 | **green** | **4050** | **480** | **4530** |
| 2 | 465 | 75 | 540 | 27 | 456 | 54 | 510 |
| 3 | 170 | 40 | 210 | 28 | 510 | 30 | 540 |
| 4 | 162 | 48 | 210 | 29 | 526 | 44 | 570 |
| 5 | 14 | 16 | 30 | 30 | 478 | 62 | 540 |
| 6 | 85 | 35 | 120 | 31 | 499 | 71 | 570 |
| **yellow** | **1698** | **222** | **1920** | 32 | 421 | 29 | 450 |
| 9 | 366 | 24 | 390 | 33 | 446 | 94 | 540 |
| 10 | 438 | 42 | 480 | 34 | 386 | 64 | 450 |
| 11 | 454 | 116 | 570 | 35 | 328 | 32 | 360 |
| 12 | 232 | 8 | 240 | **blue-green** | **4114** | **26** | **4140** |
| 13 | 156 | 24 | 180 | 36 | 443 | 7 | 450 |
| 14 | 22 | 8 | 30 | 37 | 447 | 3 | 450 |
| 15 | 30 | 0 | 30 | 38 | 596 | 4 | 600 |
| **yellow-green** | **2953** | **257** | **3210** | 39 | 478 | 2 | 480 |
| 18 | 421 | 29 | 450 | 40 | 508 | 2 | 510 |
| 19 | 454 | 26 | 480 | 41 | 449 | 1 | 450 |
| 20 | 467 | 73 | 540 | 42 | 415 | 5 | 420 |
| 21 | 344 | 16 | 360 | 43 | 449 | 1 | 450 |
| 22 | 374 | 46 | 420 | 44 | 329 | 1 | 330 |
| 23 | 273 | 27 | 300 | | | | |
| 24 | 231 | 9 | 240 | **Total** | **14231** | **1309** | **15540** |

$\chi^2 = 954.020$, df = 40, $p = 0.000^{**}$

**Figure A5.** Results of the Chi-square test for color recognition accuracy among the different colors for the interior forest landscape images.

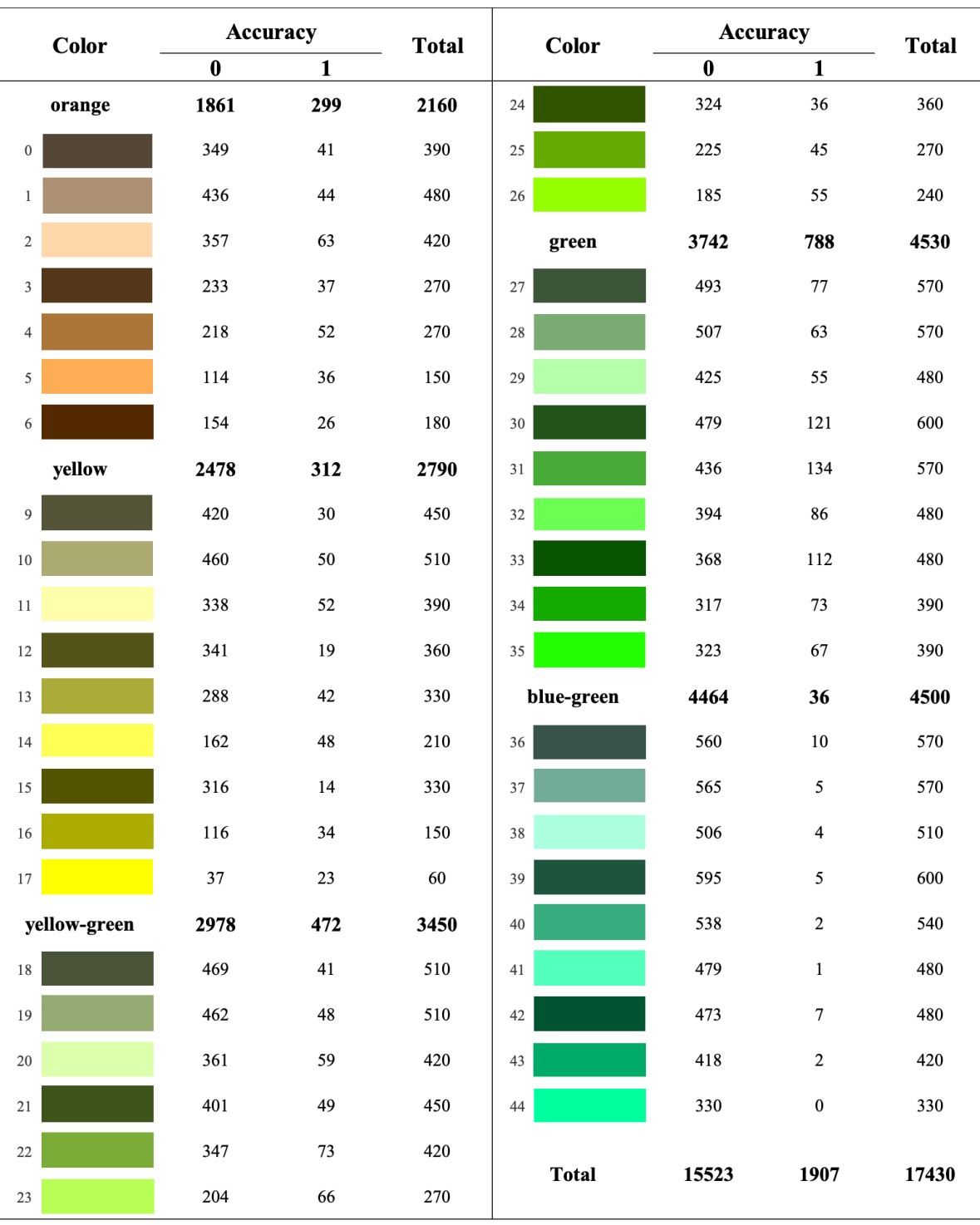

| Color | Accuracy | | Total | Color | Accuracy | | Total |
|---|---|---|---|---|---|---|---|
| | **0** | **1** | | | **0** | **1** | |
| **orange** | **1861** | **299** | **2160** | 24 | 324 | 36 | 360 |
| 0 | 349 | 41 | 390 | 25 | 225 | 45 | 270 |
| 1 | 436 | 44 | 480 | 26 | 185 | 55 | 240 |
| 2 | 357 | 63 | 420 | **green** | **3742** | **788** | **4530** |
| 3 | 233 | 37 | 270 | 27 | 493 | 77 | 570 |
| 4 | 218 | 52 | 270 | 28 | 507 | 63 | 570 |
| 5 | 114 | 36 | 150 | 29 | 425 | 55 | 480 |
| 6 | 154 | 26 | 180 | 30 | 479 | 121 | 600 |
| **yellow** | **2478** | **312** | **2790** | 31 | 436 | 134 | 570 |
| 9 | 420 | 30 | 450 | 32 | 394 | 86 | 480 |
| 10 | 460 | 50 | 510 | 33 | 368 | 112 | 480 |
| 11 | 338 | 52 | 390 | 34 | 317 | 73 | 390 |
| 12 | 341 | 19 | 360 | 35 | 323 | 67 | 390 |
| 13 | 288 | 42 | 330 | **blue-green** | **4464** | **36** | **4500** |
| 14 | 162 | 48 | 210 | 36 | 560 | 10 | 570 |
| 15 | 316 | 14 | 330 | 37 | 565 | 5 | 570 |
| 16 | 116 | 34 | 150 | 38 | 506 | 4 | 510 |
| 17 | 37 | 23 | 60 | 39 | 595 | 5 | 600 |
| **yellow-green** | **2978** | **472** | **3450** | 40 | 538 | 2 | 540 |
| 18 | 469 | 41 | 510 | 41 | 479 | 1 | 480 |
| 19 | 462 | 48 | 510 | 42 | 473 | 7 | 480 |
| 20 | 361 | 59 | 420 | 43 | 418 | 2 | 420 |
| 21 | 401 | 49 | 450 | 44 | 330 | 0 | 330 |
| 22 | 347 | 73 | 420 | | | | |
| 23 | 204 | 66 | 270 | **Total** | **15523** | **1907** | **17430** |

$\chi^2 = 1083.400, \quad df = 42, \quad p = 0.000^{**}$

**Figure A6.** Results of the Chi-square test for color recognition accuracy among the different colors for the distant forest landscape images.

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
