# Peer review of "Constructing a Forest Color Palette and the Effects of the Color Patch Index on Human Eye Recognition Accuracy"

_forests, doi:10.3390/f14030627_

Round 1
Reviewer 1 Report
Dear Authors,
Presented paper is orignial. Generally the whole chapyers were presented in good quality, but I need here more practical applications these results.
Please add it.
Reviewer 2 Report
The article is a preliminary theoretical study addressing colour in forest parks of China.
The manuscript is correctly structured and referenced.
I have few suggestions which may help to improve the paper.
The lines 94-104 would fit better into the Methods section while lines 162-167 into the Results.
To highlight the authors’ own interpretation of the results and their impact on the forest park policy it would be useful to put the lines 435-441, 451-457 and 475-477 into a separate section “Guidelines’ e. g. following the Conclusions.
Reviewer 3 Report
This is an interesting paper on forest colors. It is very interesting result that the recognition accuracy was the lowest for blue-green, although the recognition accuracy was the highest for green. Please consider the following points.
1. It has been reported that the number of color patches, edge density, splitting index, and cohesion have a significant negative effect. What are the possible reasons for this? Please add further consideration.
2. It is stated that it is necessary to use appropriate algorithms to quantify color information in the future. What theories and techniques are there for that "good algorithm"? Please add an explanation.
Reviewer 4 Report
The article is a well-prepared research paper addressing a topic important from the perspective of forest landscape planning and design. Although it focuses only on color as the most important visual element and neglects the role of non-visual stimuli, I find the work valuable, making an important contribution to the study of the perception of forest landscapes. However, I propose to supplement the paper with some elements to strengthen the scientific considerations presented in it.
1. in the abstract, I propose to write what is the purpose of the work
2. in the keywords, I propose to add the word perception or vision and China
3. in the introduction, I suggest to add content on the perception of forest landscapes and highlight the research of C. Song et al. (2018), Physiological Effect of Visual Stimulation with Forest Imaginery...and M. Horiuchi et al. (2014), Impact of Viewing vs. Not Viewing a Real Forest...In addition, you should write what the purpose of the paper is; admittedly, there are research questions, but in my opinion, you need to clearly state what the purpose of the research was. The third of the questions, in my opinion, needs to be made more specific, because I think the Authors were not referring to broad factors only effect of forest color patch indices on human color recognition accuracy
4. in chapter 2.1. I propose to add a figure (map) with the location of selected 40 forest parks
5. to section 2.3.2. I propose to add as an appendix a form of informed consent of the participants of the experiment
6. in the discussion, also refer to the publications mentioned above and reference to the research limitations
7. in the conclusions, please add the word "forest" in verse 497 before the phrase "landscape planning research"; it is also worth expanding the thought presented in the sentence "This provides new insights for forest color quantification and landscape planning research" (verse 497-498).
